# Drivers of the spatial phytoplankton gradient in estuarine-coastal systems: generic implications of a case study in a Dutch tidal bay

Long Jiang[1,2,3], Theo Gerkema[3], Jacco C. Kromkamp[3], Daphne van der Wal[3,4], Pedro Manuel Carrasco De La Cruz[5], and Karline Soetaert[3]

[1] Key Laboratory of Marine Hazards Forecasting, Ministry of Natural Resources, Hohai University, Nanjing, China
[2] College of Oceanography, Hohai University, Nanjing, China
[3] NIOZ Royal Netherlands Institute for Sea Research, Department of Estuarine and Delta Systems, and Utrecht University, P.O. Box 140, 4400 AC Yerseke, The Netherlands.
[4] Faculty of Geo-Information Science and Earth Observation (ITC), University of Twente, P.O. Box 217, 7500 AE, Enschede, Netherlands.
[5] NF-POGO Centre of Excellence, Alfred Wegener Institute, Kurpromenade 201, 27498 Helgoland, Germany.

*Correspondence to*: Long Jiang (ljiang@hhu.edu.cn)

**Abstract.** As the primary energy and carbon source in aquatic food webs, phytoplankton generally display spatial heterogeneity due to the complicated biotic and abiotic controls, but our understanding of its causes is challenging as it involves multiple regulatory mechanisms. We applied a combination of field observation, numerical modeling, and remote sensing to display and interpret the spatial gradient of phytoplankton biomass in a Dutch tidal bay (the Oosterschelde) on the east coast of the North Sea. The 19-year (1995–2013) monitoring data reveal a seaward increasing trend in chlorophyll a concentrations during the spring bloom. Using a calibrated and validated three-dimensional hydrodynamic-biogeochemical model, two idealized model scenarios were run, switching off the suspension feeders and halving the open-boundary nutrient and phytoplankton loading. Results reveal that bivalve grazing exerts a dominant control on phytoplankton in the bay and that the tidal import mainly influences algal biomass near the mouth. Satellite data captured a post-bloom snapshot indicating the temporally variable phytoplankton distribution. Based on a literature review, we found five common spatial phytoplankton patterns in global estuarine-coastal ecosystems in comparison with the Oosterschelde case: seawards increasing, seawards decreasing, concave with a chlorophyll maximum, weak spatial gradients, and irregular patterns. We highlight the temporal variability of these spatial patterns and the importance of anthropogenic and environmental influences.

## 1 Introduction

As the most important energy source in aquatic systems, phytoplankton account for 1% of the global biomass but create around 50% of the global primary production (Boyce et al., 2010). Located at the land-ocean interface, estuarine-coastal systems, including estuaries, bays, lagoons, fjords, river deltas, and plumes, are relatively productive and abundant in phytoplankton (Carstensen et al., 2015). As the basis of the pelagic food web, phytoplankton have an immense impact on the biogeochemical cycles, water quality, and ecosystem services (Cloern et al., 2014). A sound understanding of the spatial

variability of phytoplankton is critical for effective assessment, exploitation, and protection of estuarine-coastal ecosystems but remains a challenge due to the complicated natural and anthropogenic controls (Grangeré et al., 2010; Srichandan et al., 2015).

The standing stock of phytoplankton is a function of sources and sinks that are subject to both biotic and abiotic influences (Lancelot and Muylaert, 2011; Jiang et al., 2015). Phytoplankton growth is regulated by bottom-up factors such as nutrients, light, and temperature (Underwood and Kromkamp, 1999; Cloern et al., 2014), while natural mortality and grazing pressure from zooplankton, suspension feeders, and other herbivores contribute to the loss of phytoplankton biomass (Kimmerer and Thompson, 2014). Physical transport can act as either a direct source or sink, driving algal cells into or out of a certain region (Martin et al., 2007; Qin and Shen, 2017). The hydrodynamic conditions also affect the phytoplankton biomass indirectly. For example, phytoplankton growth is dependent on transport of dissolved nutrients (Ahel et al., 1996). Tides and waves affect concentrations of light-shading suspended particulate matter (SPM) and thus photosynthesis (Soetaert et al., 1994). The efficiency of benthic filtration feeding on the surface phytoplankton is associated with stratification of the water column (Hily, 1991; Lucas et al., 2016).

For these reasons, the phytoplankton distribution in estuarine-coastal systems relies on the spatial patterns of physical, chemical, and biological environmental factors of each system (Grangeré et al., 2010). For example, phytoplankton variability in one semi-enclosed water body can be dominated by terrestrial input (river-dominated), oceanic input (tide-dominated), top-down effects (grazing-dominated), others, or a combination of the above factors. More often, it is the delicate balance of multiple factors that determine phytoplankton gradients. Under high river discharge, phytoplankton growth can be promoted by increasing nutrient input, whereas advective loss and high riverine SPM loading may inhibit algal enrichment (Lancelot and Muylaert, 2011; Shen et al., 2019). In tide-dominated systems, tides can resuspend SPM, negatively impacting phytoplankton, while at the same time bringing regenerated nutrients into the water column, or drive upwelling-induced algal blooms from the coastal ocean into estuaries (Sin et al., 1999; Roegner et al., 2002). Nitrate can support more phytoplankton biomass in microtidal estuaries than in macrotidal estuaries (Monbet, 1992). The relative importance of zooplankton and bivalve grazing on phytoplankton varies spatially (Kromkamp et al., 1995; Herman et al., 1999; Kimmerer and Thompson, 2014). These complexities make it challenging to discern the driving mechanisms of the spatial phytoplankton gradient, and comparative studies of different systems are lacking (Kromkamp and van Engeland, 2010; Cloern et al., 2017).

*In situ* observation, remote sensing, and numerical modeling are common techniques to reveal spatial patterns and detect their biophysical controls (Banas et al., 2007; Grangeré et al., 2010; Srichandan et al., 2015). Shipboard measurements of chlorophyll a (chl-a) provide a precise and dynamic assessment of the phytoplankton variability; however, the temporal (usually monthly) and spatial (usually tens of kilometers) resolutions are limited compared to satellite images and numerical models (Soetaert et al., 2006; Valdes-Weaver et al., 2006; van der Molen and Perissinotto, 2011; Cloern and Jassby, 2012; Kaufman et al., 2017). Remote sensing of chl-a reveals the surface distribution with a sufficiently high spatial resolution and coverage, but only at favorable weather conditions (Srichandan et al., 2015). In comparison, ecological models are based on

simplified assumptions and numerical formulations and cannot simulate every detail of natural processes. However, a properly calibrated and validated model is capable of representing the system of interest at a fine resolution (Friedrichs et al., 2018) and allows testing hypotheses of mechanisms driving the phytoplankton distribution (Jiang and Xia, 2017, 2018; Irby et al., 2018). As a reliable biophysical model must be based on observational and satellite data (Soetaert et al, 1994; Feng et al., 2015; Jiang et al., 2015), a combination of these approaches is optimal to improve our knowledge of the spatial heterogeneity in estuarine-coastal ecosystems.

In this study, we combined satellite, long-term monitoring, and numerical modeling to investigate the potential drivers of the spatial phytoplankton gradients in a well-mixed tidal bay, the Oosterschelde (SW Netherlands). In this case study, we identified the main environmental drivers of spatial phytoplankton distribution in the bay, and used some sensitivity model tests to quantify the impact of these drivers. Through such a mechanistic investigation into the spatial phytoplankton gradient, our case study was then used as a prototype in comparing spatial phytoplankton gradients among global estuaries and coastal bays. Based on a literature review, five main types of spatial heterogeneity in phytoplankton biomass are identified along with examples and dominant controls.

## 2 The study site

The Oosterschelde is located in the Southwest Delta of the Netherlands (Fig. 1). Due to the flood-protection constructions named Delta Works since the 1980s, the delta region has changed from an interconnected water network to individual water basins isolated by dams and sluices (Ysebaert et al., 2016). The confluence of the Rhine and Meuse Rivers flow into the North Sea through a narrow channel (Fig. 1) with a combined discharge of over 2000 $m^3$/s (Ysebaert et al., 2016). The Westerschelde (Fig. 1) is the only remaining estuary in the delta region covering fresh to saline waters (Ysebaert et al., 2016). With the substantially reduced freshwater input, the Oosterschelde has been filled with saline water (salinity 30–33), lost characteristics of an estuary, and developed into a tidal bay (Nienhuis and Smaal, 1994; Wetsteyn and Kromkamp, 1994). The northernmost end in the northern branch has salinities fluctuating between 28.5 and 30.5, caused by small freshwater inflow through the Krammer sluice. However, the occasional freshwater flux controlled by the sluice is mostly below 10 $m^3$/s (Ysebaert et al., 2016). This is negligible compared to the tidal exchange, which is ~2 × $10^4$ $m^3$/s, estimated from a typical tidal prism of 9 × $10^8$ $m^3$ in a 12-h tidal cycle. The tidal prism is about one third of the basin volume, ~2.76× $10^9$ $m^3$ (Nienhuis and Smaal, 1994). As part of the Delta Works, a partially-open storm surge barrier was implemented at the mouth of the Oosterschelde, which is occasionally closed during severe storms. Since then, the tidal basin still experiences a semi-diurnal tidal regime, but the average tidal range has been reduced by ~13% to 2.5–3.4 m from the west to east, the tidal flat area was reduced, and current velocity decreased (Nienhuis and Smaal, 1994; Vroon, 1994). In the post-barrier decades, the entire basin has been dominated by the tidal exchange with the North Sea, causing net import of phytoplankton biomass and seston; the water residence time (RT) of the bay ranges 0–150 days from the western to eastern ends (Jiang et al., 2019a).

The phytoplankton composition in the Oosterschelde has also changed since the Delta Works: the previously dominating diatoms have decreased, while the small flagellates and weakly silicified diatoms became more abundant, especially in summer (Bakker et al., 1994). The annual cycle of phytoplankton biomass is characterized by a spring bloom and a much weaker late summer peak (Wetsteyn and Kromkamp, 1994). The Oosterschelde is extensively used for aquaculture of Pacific oysters (*Crassostrea gigas*) and blue mussels (*Mytilus edulis*) in the past decades, and their annual productions are approximately 3 and 20–40 kt fresh weight, respectively (Smaal et al., 2009; Wijsman et al., 2019). Oysters, mussels, and wild cockles (*Cerastoderma edule*) are the main benthic suspension feeders in the basin (Fig. 1). Strong pelagic-benthic coupling has been reported for the Oosterschelde ecosystem: benthic filtration very likely accounts for the declining annual primary production (Smaal et al., 2013). In addition, abundant benthic suspension feeders make the Oosterschelde an important feeding ground and international conservation zone for wading birds (Tangelder et al., 2012).

## 3 Methods

### 3.1 Field observations

From 1995 to 2013, the Royal Netherlands Institute for Sea Research (NIOZ) conducted shipboard monitoring of the Oosterschelde on a biweekly to monthly basis. The monitoring campaign routinely collected water samples at eight stations in the basin (OS1–OS8, Fig. 2) for nutrient measurements and filtered them for measurements of chl-a and SPM. The dataset has been applied in several previous studies (Cloern and Jassby, 2010; Smaal et al., 2013), and the sampling method was the same as described by Kromkamp and van Engeland (2010). The Dutch government agency Rijkswaterstaat (RWS) has monitored nutrients and chl-a in the Oosterschelde at different locations (e.g., RWS1–RWS4, Fig. 2), and these monthly data are freely accessible on the RWS data portal (https://waterinfo.rws.nl). Compared with the NIOZ data, the RWS data include two offshore stations RWS1 and RWS2 (Fig. 2). Since the study region is mostly well-mixed (Wetsteyn and Kromkamp, 1994), both datasets used surface samples to represent the water column at each station.

Primary production was estimated by $^{14}CO_2$ uptake (mg-C h$^{-1}$) during two-hour incubation experiments for all 16 NIOZ sampling dates in 2010 at stations OS2 and OS8 (Fig. 2). Incubation experiments were conducted following a previously described method (Kromkamp and Peene, 1995; Kromkamp et al., 1995). A *PI*-curve linking irradiance ($I$, µmol-photons m$^{-2}$ s$^{-1}$) to the chl-a normalized C-fixation rate ($P$, mg-C mg-chl-a$^{-1}$ h$^{-1}$) was mathematically represented by a maximum C-fixation rate ($P_m$, mg-C mg-chl-a$^{-1}$ h$^{-1}$), an initial slope of the curve ($\alpha$, mg-C mg-chl-a$^{-1}$ h$^{-1}$/µmol-photons m$^{-2}$ s$^{-1}$) and the irradiance at which $P_m$ occurred ($I_{opt}$, µmol-photons m$^{-2}$ s$^{-1}$) according to the model of Eilers and Peeters (1988). Light intensity at multiple water depths was measured in the field with Licor LI-192SB cosine-corrected light sensors connected to a Licor LI-185B quantum meter to estimate the light extinction coefficient $K_d$ (m$^{-1}$) and generate the light attenuation curve $I_z = I_0 \exp(-zK_d)$, where $I_0$ and $z$ are surface light intensity and water depth, respectively. With the hourly photosynthetically active radiation (PAR) measured at the NIOZ as $I_0$, the hourly PAR ($I_z$) throughout the water column was computed. For a full description, see Kromkamp and Peene (1995). Then, with $P_m$, $\alpha$, $I_{opt}$, and $I$ available, the hourly

photosynthetic rate at each water depth ($P_z$) was calculated and integrated over depth to obtain the primary production of the entire water column and during the whole day. We used the measured primary production data without estimating the respiratory losses as respiration will not affect the N-content of the algae. In a short incubation time, the [14]C method is often thought to reflect gross primary productivity (*GPP*). However, results by Halsey et al. (2010, 2013) showed that even a 30 min [14]C incubation experiment can reflect *GPP* at low growth rates and net primary productivity (*NPP*) at high growth rates. Hence, as during the main growing seasons growth rates are generally high (Underwood and Kromkamp, 1999), we assumed that our [14]C-method reflected *NPP* measurements. The phytoplankton turnover time (PT) was calculated by dividing the observed phytoplankton biomass by *NPP*.

## 3.2 Numerical modeling

A three-dimensional hydrodynamic-biogeochemical model GETM-FABM (General Estuarine Transport Model coupled with the Framework for Aquatic Biogeochemical Models) was applied in a two-year (2009–2010) simulation to identify drivers of spatial phytoplankton dynamics in the Oosterschelde. GETM and FABM are open-source models, available from websites https://getm.eu/ and https://github.com/fabm-model/fabm. The model was implemented on a 300 m × 300 m rectangular grid with 10 equally-divided vertical layers, covering the Oosterschelde and part of the North Sea (Fig. 1). The hydrodynamic model using GETM version 2.5.0 was driven by realistic meteorological forcing (winds, irradiance, air pressure, etc.) and tides and the output water level, temperature, salinity, and current velocity were calibrated and validated with observational data (Jiang et al., 2019a). Jiang et al. (2019a) provided a detailed description of the GETM setup and model validation. The validation of FABM is presented in Section 4.2.

The biogeochemical model was coupled online with GETM on the FABM platform (Bruggeman and Bolding, 2014). The physical and biogeochemical simulations were conducted simultaneously with a time step of 8 s. In each time step, GETM provided FABM with the environmental variables, such as temperature, water elevation, and irradiance. The transport and mixing of nutrients, detritus, and plankton biomass was represented by the same equation as that of salinity except that phytoplankton and detritus sank at a speed of 0.2 m d$^{-1}$ and 1.0 m d$^{-1}$, respectively (Eppley et al., 1967; Soetaert et al., 2001).

Our biogeochemical model was nitrogen-based and consisted of a pelagic and benthic module (Fig. 3). The pelagic module was a typical NPZD framework comprising the state variables Nutrient (DIN, dissolved inorganic nitrogen), Phytoplankton, Zooplankton, and Detritus (unit: mmol nitrogen m$^{-3}$), while the benthic variables in mmol nitrogen m$^{-2}$ included benthic detritus, microphytobenthos, and the three dominant bivalve species in the Oosterschelde: mussels, oysters and cockles. All mass transfer processes are listed as arrows in Fig. 3, and the main formulations, variables, and parameters for calculating these processes are summarized in Tables 1 and 2. The climatological data in December and January were averaged using the 19-year observations and were used as the initial model condition. The shellfish distribution (see Fig. 1) and annual biomass in 2009 and 2010 were estimated by Wageningen Marine Research (Smaal et al., 2013; Jiang et al., 2019a). The model output was compared with available observational DIN, chl-a, and *NPP* described in Section 3.1. Given

that *NPP* was measured by carbon-based methods, the nitrogen-based simulation results were converted to carbon using the Redfield ratio (C:N = 6.625 mol C mol N$^{-1}$). Phytoplankton biomass was measured in Chlorophyll units. We prescribed a Chl:N ratio (Chl:N = 2 mg Chl mmol N$^{-1}$, Soetaert et al., 2001) to compare our model output to the chlorophyll data.

In order to investigate the roles of coastal influx and benthic grazing in shaping the spatial phytoplankton patterns in the basin, we conducted two idealized numerical scenarios in addition to the realistic (baseline) run. One scenario was halving the DIN concentration and phytoplankton biomass at the open boundary (i.e., the North Sea including Westerschelde and Rhine river plumes, see Fig. 1). The other scenario switched off the bivalve state variables. Based on our assessment, the effects of freshwater input on DIN and chl-a were minimal, local, and far less significant than the above two factors. Thus, the sensitivity runs of the freshwater input will not be elaborated hereafter.

### 3.3. Satellite remote sensing

A clear sky Sentinel-2 MSI (10 m spatial resolution) satellite image of 11 May 2018 (10:55 UTC) for tile 31UET was downloaded as level 1C data from the Copernicus Sentinel hub (https://scihub.copernicus.eu). The Acolite processor (version Python 20190326.0) developed by RBINS (Vanhellemont and Ruddick, 2016) was applied using default settings to correct for atmospheric (aerosol) effects based on a dark spectrum fitting (Vanhellemont and Ruddick, 2018; Vanhellemont, 2019), to flag clouds and land, and to retrieve chl-a concentrations, using the red edge algorithm defined by Gons et al. (2002) with a mass specific chl-a absorption set to 0.015. Data were extracted in the Sentinel Application Platform (SNAP version 7.0.0) and converted to a GeoTIFF raster for further processing in ArcGIS. Georeferencing of the raster was enhanced using an affine transformation to a detailed topographic map of Rijkswaterstaat. The satellite image was acquired during high water: water level at Rijkswaterstaat tide gauge station Stavenisse (https://waterinfo.rws.nl/%20/nav/index) was +1.12 m NAP incoming tide during overpass. A Sentinel-2 MSI image of 21 April 2019 (10:56 UTC) was acquired during low water conditions (such as -1.58m NAP incoming tide), and processed in the same way. "Land" flags obtained from this low water image were used to further flag shallow waters (i.e., the inundated tidal flats) in the highwater image, to avoid potential bottom reflectance.

### 4 Results

### 4.1 Field observations

The 19-year chl-a time series illustrates the seasonal pattern of phytoplankton biomass in the Oosterschelde (Fig. 4). The spring bloom takes place in March or April during conditions of increasingly favorable temperature, light, and nutrients and lasts less than a month. The peak biomass varies dramatically interannually (Fig. 5), with smaller peaks during different months, especially in 2010 (Fig. 4). Likely due to nutrient limitation and grazing pressure, the summer biomass stays mostly below 10 mg m$^{-3}$ (Figs. 4 and 5). In the well-mixed Oosterschelde with limited freshwater input, temperature and light

constrain algal growth in winter, when nutrients accumulate and phytoplankton biomass falls below 3 mg m$^{-3}$ (Figs. 4 and 5). These seasonal controls of phytoplankton variability are substantiated with the numerical model in Section 4.2.

To better display the spatial chl-a gradients, monthly averages and standard deviations of the 19-year observations are displayed (Fig. 5). A decreasing chl-a gradient from the mouth (OS1) to head (OS8) of the basin is observed mainly during the spring bloom, whereas the spatial phytoplankton gradient is not as pronounced in summer and winter (Fig. 5a). The station RWS1 that is close to the mouth of the Westerschelde estuary usually has a higher chl-a concentration than further offshore (RWS2) and in the Oosterschelde (RWS3 and RWS4) (Fig. 5b). Despite interannual variability in the timing of the bloom and different sampling time every year, the period March to May mostly covers the initiation, development, and wane of the spring bloom. The 19-year average phytoplankton biomass during this season demonstrates a clear gradient in the bay and adjacent coastal sea (Fig. 2). The chl-a decreases from the Westerschelde plume region (RWS1) offshore (RWS2) and further into the eastern and northern ends of the Oosterschelde (Fig. 2).

## 4.2 Numerical modeling

The model results compared to observed concentrations of DIN and chl-a in a two-year simulation are shown in Fig. 6. Most DIN consumption happens during the spring bloom, and the regenerated DIN accumulates over winter until the next bloom sets off. The simulated chl-a during the bloom demonstrates the same gradient between the western and eastern bay as observed (OS1 > OS3 > OS8, Figs. 6d–6f). The model skill is quantified by $r^2$ ($r^2 = 0.89$ for DIN and 0.66 for chl-a) and root mean square errors ($RMSE = 6.0$ mmol m$^{-3}$ for DIN and 3.9 mg m$^{-3}$ for chl-a) between simulation and observation. Despite capturing the major seasonal and spatial patterns, the model seems to miss some details such as overestimating the recycled DIN at OS8 and showing a slower collapse of spring blooms than observed. Meanwhile, the daily time series of the model output exhibits spring-neap biweekly fluctuations (Fig. 6) that cannot be substantiated by the observations owing to a low sampling frequency.

The modeled *NPP*, the product of phytoplankton biomass and growth rate, is in general agreement with the measurements (Fig. 7, black line). According to Equation (2) in Table 1, the growth rate is a function of temperature, nutrient, and light factors in the model. Here we decompose the seasonal cycle of these three factors and use their product to assess the growth rate (Fig. 8). Before the bloom, both modeled phytoplankton biomass and growth rates are low, resulting in a low *NPP* (Figs. 7 and 8). The fast-growing period, around Day 100 as a consequence of increased temperature and light (Figs. 7 and 8), triggers the increase in simulated biomass that leads to the bloom. The spring bloom is terminated due to enhanced nutrient limitation around Day 125 in the model (Figs. 7 and 8). In the low-biomass post-bloom summer (Fig. 6), both modeled and measured *NPP* is only slightly lower than that in the bloom (Fig. 7) and the environmental conditions are still favorable for growing (Fig. 8). The summer growth rate is mainly fueled by regenerated nutrients, while the low biomass results substantially from grazing. The model underestimated DIN and thus *NPP* after the spring bloom (Days 490–540 in Fig. 6 and 125–175 in Fig. 7) and overestimated the recycled DIN at OS8 in fall 2010 (Days 600–650 in Fig. 6a), which explains the overestimation of *NPP* in this period (Days 235–285 in Fig. 7a). The observed and simulated *NPP* at OS8

(902.6 ± 928.4 mg C m$^{-2}$ d$^{-1}$ and 1033.9 ± 1084.3 mg C m$^{-2}$ d$^{-1}$, respectively) is generally higher than that at OS2 (722.5 ± 794.6 mg C m$^{-2}$ d$^{-1}$ and 606.0 ± 499.5 mg C m$^{-2}$ d$^{-1}$, respectively). According to the one-tail t-test, the difference between observed *NPP* at these two stations is not significant ($t = 0.59$, $p > 0.05$, $n = 16$), whereas due to the overestimation the simulated *NPP* at OS 8 is significantly higher than that at OS2 ($t = 6.85$, $p < 0.05$, $n = 365$). This is in contrast to the chl-a, which is higher at OS2 (Fig. 2). Based on the observed chl-a and measured *NPP*, PT during the warm months (March to October) is 0.92–5.2 days and 0.13–4.3 days at OS2 and OS8, respectively.

The calibrated and validated model was used to map the 15-day average chl-a during the peak bloom in 2009 (Fig. 9). The North Sea exhibits significantly higher algal biomass than the Oosterschelde (Fig. 9). Inside the bay, phytoplankton biomass is clearly low over the shellfish-colonized area (compare Figs. 1 and 9). The north-south and east-west chl-a gradients observed in field monitoring data are reproduced in the model results (Figs. 2 and 9).

When switching off bivalve activities, the modeled phytoplankton biomass significantly increases, especially at the eastern station OS8 (Fig. 10). At this station, the chl-a during the bloom is nearly tripled, it doubled at OS3 and increased by 20% at OS1, respectively (Fig. 10). The west to east spatial chl-a gradient is weakened in spring and even reversed in summer, i.e., concentrations decrease seawards (Fig. 10). Remarkably, without bivalves, the summer *NPP* at OS8 is not greatly affected (Fig. 7a) despite increased algal biomass, which implies a reduction in the growth rate (Equation (2), Table 1). Given the unchanged light and temperature in the no-bivalve scenario, the reduced growth rate results from diminished nutrient regeneration. The summer *NPP* at OS2 is increased when bivalves are turned off (Fig. 7b), which is a consequence of increased phytoplankton biomass.

Halving the DIN and phytoplankton loading from the North Sea hardly has an influence on the *NPP* in the Oosterschelde model (Fig. 7). This indicates that allochthonous coastal nutrients are not a major source of inner-bay primary production, which relies mainly on recycling. With halved coastal import, the modeled peak phytoplankton biomass is nearly halved at OS1, but the reduction is lower at OS3 (~35%) and OS8 (~20%) (Fig. 10). Therefore, tidal import has its modeled impact mainly exclusively near the bay mouth during the bloom. This contrasts to the benthic bivalves that appear to exert grazing pressure all over the bay and stimulates primary production by replenishing inorganic nutrients into the water column, the latter process being crucial in nutrient-depleted seasons.

## 4.3 Satellite remote sensing

Remote sensing images with sufficient spatial resolution, in this case the Sentinel-2 MSI data, are utilized to complement the spatial patterns shown in observational and modeling data. In an attempt to find images during the spring bloom and high tide (to avoid interference from bottom reflectance), we only found one post-bloom snapshot under clear sky (Fig. 11a). This provides additional insight into the observed and modeled spatial chl-a pattern. On 11 May 2018, the chl-a concentration was highest in the central basin and reduced eastwards and northwards into the highly bivalve-populated areas (Figs. 1 and 11a), consistent with the chl-a gradient described in Sections 4.1 and 4.2. However, the post-bloom chl-a concentration was low in the North Sea so that low import was shown in the southwestern bay near the mouth (Fig. 11a).

Such a spatial chl-a pattern with higher concentrations in the central basin (Fig. 11a) is often present in the model results. For example, in the post-bloom period in 2008 and 2009, the chl-a concentration at OS3 is higher than OS1 and OS8 at times (Fig. 5a). Likewise, in a post-bloom model snapshot during high tide on 1 May 2010 (Fig. 11b), the phytoplankton distribution exhibits a similar spatial gradient as in Fig. 11a ($r^2 = 0.118$, $n = 2817$, $p < 0.0001$).

## 5 Discussion

The approaches applied in this case study including field observation, numerical modeling, and satellite remote sensing each have their drawbacks. The monitoring data is not frequent enough to capture the peak bloom that lasts only a couple of weeks and misses details in spatial distribution between stations. The temporal resolution of the satellite data is even lower, but the spatial detail is very high. The model, while resolving spatial and temporal scales at a high resolution, is based on simplified assumptions. The NPZD model considers nitrogen only and assumes no phosphorus or silicon limitation in phytoplankton growth. In late spring, phosphorus or silicon may become limited in the Oosterschelde (Wetsteyn and Kromkamp, 1994; Smaal et al., 2013). This likely explains the faster DIN consumption in the simulated data compared to the observation (Figs. 6a–6c), resulting in the accelerated nitrogen limitation and underestimation of the post-bloom $NPP$ (Figs. 7 and 8). Additionally, our model does not account for the shellfish harvest, mostly in late summer, which can contribute to the overestimation of the regenerated DIN and hence $NPP$, especially in the eastern part (e.g., Figs. 6a and 7a). When converting the nitrogen-based model results to compare with chl-a and the carbon-based $NPP$, the Chl:N and Redfield ratios were applied, without considering the variable stoichiometry in natural phytoplankton groups. A model accounting for the phytoplankton physiological plasticity (Faugeras et al., 2004), e.g., low (high) chl-a content under high (low) light intensity and high C:N ratios under nitrogen limitation, should be considered in further studies. Despite these simplifications and limitations, the approaches complement each other in the spatiotemporal resolution and coverage and offer insight into the phytoplankton distribution in the Oosterschelde, as well as the underlying mechanisms.

Grazing by filtration feeders is found to be the dominant factor shaping the spatial and seasonal phytoplankton patterns in the Oosterschelde. In the eastern and northern bay, RT is relatively long (>100 days, calculated by the e-folding method and the remnant function by Jiang et al., 2019b), water depth and cross-bay area are much smaller (Fig. 12c), and the tidal amplitude and mixing are stronger due to geometric convergence (Jiang et al., 2020), which contributes to stronger pelagic-benthic coupling and creates favorable feeding conditions for suspension feeders (Hilly, 1991). Thus, over and near the shellfish habitat, the phytoplankton biomass is usually low, even during the bloom (Figs. 1 and 9). Smaal et al. (2013) attributed the decline of the annual primary production and chl-a concentration in the Oosterschelde to overgrazing, as found in the Bay of Brest (Hilly, 1991) and many Danish estuaries (Conley et al., 2000). Our findings support the predominant top-down control on phytoplankton distribution and standing stocks (Fig. 10), as well as on primary production, particularly in the post-bloom seasons (Fig. 7b). It has been shown that a recruitment failure of mussels and cockles promotes primary production and algal accumulation in the Dutch Wadden Sea (Beukema and Cadée, 1996), consistent with our numerical

experiment removing bivalves. Although bivalves accelerate nutrient remineralization, this positive feedback on phytoplankton growth does not compensate for the grazing loss (Figs. 7, 8, and 10). Optimization of the bivalve stock size and culture locations based on these scientific insights could enhance phytoplankton proliferation and increase the shellfish carrying capacity.

The strong top-down control by shellfish can result from cultured or natural populations. In the Oosterschelde, the high shellfish biomass consists of both wild cockles and oysters, as well as commercial Pacific oysters and blue mussels (see the data in Jiang et al., 2019a). Worldwide, shellfish culture can be an important source of benthic grazing. Examples include the farmed oyster in Willapa Bay (Banas et al., 2007) and cultivated mussels in the Baie des Veys estuary (Grangeré et al., 2010). Invasive shellfish species can also exert significant grazing pressure. After their introduction, the invasive clam

*Potamocorbula amurensis* in San Francisco Bay (Lucas et al., 2016) and the exotic dreissenid mussels (*Dreissena spp.*) in the Hudson River (Strayer et al., 2008) and Laurentian Great Lakes (Higgins and Vander Zanden, 2010) quickly dominated the benthos, strongly increased the filtration capacity, thus extensively changing the lower foodweb. Also in the Oosterschelde, it has been noted that great care should be taken with regards to the invasive shellfish, such as *Ensis americanus* (Smaal et al., 2013).

Ecologists use three timescales to assess the carrying capacity of shellfish-dominated ecosystems: clearance time (CT), the time it takes for suspension feeders to filter the entire basin, RT, and PT (Dame and Prins, 1998). A ratio CT/RT < 1 suggests that the rate at which the system is replenished is outpaced by the filtration rate, and that the pelagic ecosystem is controlled by benthic grazing. The ratio CT/PT < 1 reveals that the food (phytoplankton) reproduction is slower than filtration so that the system may collapse. In the Oosterschelde, the grazing pressure is immense (CT/RT < 1, CT = 19.6 days,

average RT > 30 days, Jiang et al., 2019a), while the system is still sustainable (CT/PT > 1). Thus the strong benthic filtration capacity consumes considerable pelagic production and puts high pressure on the pelagic foodweb (Smaal et al., 2013). Compared to other estuarine-coastal systems used for shellfish farming (e.g., the western Wadden Sea, Beatrix Bay, Narragansett Bay, etc.), the indices in the Oosterschelde suggest relatively high exploitation, including a high ratio of the overall shellfish biomass to basin volume, i.e., the relative shellfish density, and low CT/RT and CT/PT ratios (Jiang et al.,

2019a; Smaal and van Duren, 2019).

        Compared to benthic feeding, tidal import mainly influences the phytoplankton biomass in the southern channel near the mouth during the spring bloom. The Southern Bight of the North Sea, and more particular the water in river plumes (e.g., Westerschelde and Rhine plumes), is relatively productive, compared to other shelf seas (van der Woerd et al., 2011). The spring bloom in the Oosterschelde is usually not as strong as in the adjacent North Sea (Figs. 2, 5, and 9), so that tidal

import of phytoplankton from the North Sea sets the upper limit of the phytoplankton in the Oosterschelde (Fig. 10). However, in other seasons, phytoplankton biomass is very similar inside and outside of the bay, so that coastal import is not as high as in spring (Fig. 10). In spring, the measured PT is shorter than 5.2 days (Section 4.2), which is comparable to the RT near the bay mouth (Jiang et al., 2019b) so that the phytoplankton biomass import is of similar magnitude to local production in the southwestern Oosterschelde. The role of tidal import decreases further into the bay, as supported by the

numerical experiment shown in Fig. 12, and is minimal in the eastern bay (e.g, at OS8, Fig. 10a). Thus, the effect of tidal import on the phytoplankton biomass in the Oosterschelde is subject to seasonal and spatial variability, and depends on two conditions: (1) significantly different phytoplankton biomass in and out of the bay and (2) sufficiently short RT compared to the phytoplankton turnover time.

The western part of the Oosterschelde, defined as 0–10 km east to the bay mouth, while accounting for 45.0% of the basin volume excluding the northern branch, contains 60.0% of the phytoplankton biomass during the peak spring bloom (data in Fig. 12c). The impact of halving tidal import in the model, being mostly pronounced in the western Oosterschelde (Fig. 12c), thus has a strong influence on the overall phytoplankton standing stocks, reducing the total phytoplankton biomass by 38.5% during the peak bloom, from 58.7 t to 36.1 t.

Note that the observed seaward increasing phytoplankton biomass is not so pronounced in each of the 19 years, such as in 1997 and 2012 in Fig. 4a. The interannual variability of the spring bloom timing and magnitude is among the potential causes. In winter and early spring, the shallower and landlocked Oosterschelde may be warmed up a few days faster than the adjacent North Sea, which may result in earlier spring bloom at the landward end in some years, such as 1996, 1997, and 2005 shown in Fig. 4b. Thus, the resultant spatial chl-a distribution may differ if the sampling activity is
conducted during or between these two bloom windows. Given the relative low sampling frequency (every month or two weeks), different observational activities may change the spatial chl-a distribution (e.g., Year 2010 in Figs. 5a and 5b). Additionally, if the bloom is of similar magnitude in and out of the bay, the spatial gradient may not be as strong, such as in 2012 in Fig. 4a. Therefore, in consideration of the interannual variability of the spring blooms and the possible mismatch with sampling campaigns, the observational spatial phytoplankton gradient may be inconspicuous in certain years but
evident over the long term (Figs. 2 and 5).

## 6 Synthesis

The Oosterschelde represents a land-ocean transitional system that is shallow, dynamic, and driven by pelagic-benthic coupling and exchange with the sea. The grazing pressure increases into the bay because of reduced water depth and increasing bivalve biomass, RT, tidal mixing, and thus pelagic-benthic coupling, while the North Sea with higher
phytoplankton biomass is a phytoplankton source. As a result, the phytoplankton concentration during the spring bloom consistently declines from the seaward to the landward end. When halving the nutrient and algal loading from the North Sea, the phytoplankton gradient in spring is not as pronounced, although still decreasing toward the landward end (Fig. 12). Without the grazing sink however, the phytoplankton distribution tends to be spatially uniform (Fig. 12). Given the temporal variability of dominant environmental factors, the phytoplankton gradient also changes over time. In the post-bloom period
for instance, chl-a may exhibit a central maximum, or it may exhibit a constant concentration in winter. This shows that the spatial gradient of phytoplankton biomass in estuarine-coastal systems depends on the relative importance of the main

drivers of phytoplankton accumulation. Therefore, we compare the Oosterschelde case with other estuarine-coastal systems, including different spatial phytoplankton patterns and their reported dominant environmental drivers (Fig. 13).

Increasing phytoplankton biomass from the landward to seaward ends (Fig. 13) can often be ascribed to an increasing source from the seaside, an increasing sink landwards, or more favorable growth conditions towards the sea. The Oosterschelde is a typical example during the spring bloom (Fig. 8), shaped by both marine input and increasing benthic filtration landward. The seawards increasing gradient is also common in estuaries and bays open to coastal upwelling zones (e.g., the Rías Baixas of Galicia and Tomales Bay), where algal blooms generated during upwelling events are transported into bays via multiple physical mechanisms including tidal stirring and gravitational and wind-driven circulation (Figueiras, et al., 2002; Hickey and Banas, 2003; Martin et al., 2007). In contrast, phytoplankton in the Chilika Lagoon is mostly light-limited due to the massive riverine sediment loading. Here it is a seaward increase in water transparency that leads to increasing chl-a concentrations (Srichandan et al., 2015). Hence, similar phytoplankton gradients may be driven by distinct mechanisms in various systems.

Contrary to the Oosterschelde case, phytoplankton biomass decreases in the seaward direction in some systems (Fig. 13). A typical example is the Scheldt River and Westerschelde Estuary, a eutrophic and turbid estuary with salinity ranging 0–30 (Soetaert et al., 2006). Numerical models and field observation reveal that the chl-a concentration is highest in the tidal fresh portion, reduces sharply between salinity 5–10, and is maintained at a lower level towards the polyhaline region (Soetaert et al., 1994; Kromkamp et al., 1995). The high phytoplankton biomass in the upper reach is a result of tributary import, high nutrient levels, and lack of zooplankton grazers, whereas the increasing salinity stress on the freshwater species and grazing pressure in the mesohaline zone suppress the phytoplankton proliferation (Soetaert et al., 1994; Kromkamp and Peene, 1995; Muylaert et al., 2005). Similar seawards decreasing phytoplankton gradient is also found in many river and estuarine plumes (Fig. 13), where the nutrient gradient controls the phytoplankton distribution (Gomez et al., 2018; Jiang and Xia, 2018).

A chl-a maximum zone (CMZ) occurs in many estuaries with substantial freshwater input (Fig. 13). Taking the Chesapeake Bay as an example, the upper bay is characterized by high terrestrial sediment concentrations and strong light limitation for phytoplankton growth (Son et al., 2014). A turbidity maximum zone is located near the front of salt intrusion (North et al., 2004). Beyond this location, the CMZ appears in the middle reach (Jiang and Xia, 2017), while nitrogen limitation is constantly detected in the lower bay (Miller and Harding, 2007). The CMZ is a combined consequence of the optimal light conditions and abundant terrestrial nutrients, and the CMZ location and coverage shift with river discharge and weather (Fisher et al., 1988; Miller and Harding, 2007). In some other estuaries with a CMZ (e.g., the Neuse-Pamlico estuary and York River), owing to a narrow river channel and high discharge, the flushing rate in the upper estuary can be faster than the phytoplankton turnover rate, which, rather than light, limits phytoplankton accumulation (Sin et al., 1999; Valdes-Weaver et al., 2006). In these systems, the CMZ is always in wider reaches with sufficiently long RT (Valdes-Weaver et al., 2006).

If the transport loss is higher than the growth rate in the entire basin, the phytoplankton biomass is low everywhere and negatively correlated with the flow velocity (Fig. 13). The Hudson River estuary is one of such estuaries with high nutrient loading but low and hardly spatially variable chl-a (Howarth et al., 2000). After the colonization of the invasive zebra mussel (*Dreissena polymorpha*) in the 1990s, grazing and transport losses are two dominant sink terms maintaining a low basin-wide phytoplankton standing stock in the estuary (Strayer et al., 2008). Similarly, due to the non-indigenous *P. amurensis*, the San Francisco Bay witnessed a five-fold drop in chl-a and the suppression of zooplankton, and higher trophic levels (Cloern and Jassby, 2012; Lucas et al., 2016). The chl-a distribution has shifted from maximizing at the middle bay to spatially uniform (Cloern et al., 2017), which is generally associated with strong sink factor(s) distributed all over the system.

The dominant sink (or source) factor is not always distributed uniformly nor does it follow consistent gradients in estuarine-coastal systems, generating irregular phytoplankton distribution (Fig. 13). For instance, in the Baie des Veys estuary, benthic grazing by cultivated oysters results in an area of low chl-a concentrations over the oyster bed, and this patch of low chl-a is imposed onto a seawards decreasing chl-a gradient, forming an irregular spatial pattern (Grangeré et al., 2010). In the Krka estuary, an untreated sewage discharge acts as a DIN point source, increasing the phytoplankton production downstream. Without the point source, phytoplankton seem to decrease seawards (Ahel et al., 1996). In the St. Lucia estuary, controls of primary production include nutrient stoichiometry, temperature, irradiance, and hydrological changes which all vary in different sub-regions and render complex spatial heterogeneity in phytoplankton distribution (van der Molen and Perissinotto, 2011).

## 7 Summary

In the Oosterschelde, a tidal bay along the North Sea, we detect a seaward increasing phytoplankton gradient in spring in the two-decade monitoring data. This spatial chl-a pattern was also reproduced with a nitrogen-based NPZD model calibrated and verified with observational data. In an effort to understand the main drivers of such a phytoplankton gradient, two experimental model runs were performed: switching off bivalve filtration and halving the nutrient and phytoplankton concentrations in the North Sea boundary, respectively. Results indicate that the landward increasing benthic grazing pressure is the primary cause of the spatial phytoplankton gradient, while import from the North Sea tends to strengthen the gradient. The satellite image implies that tidal import is mainly influential in the southwestern bay. With the variation of these two drivers, the spatial phytoplankton distribution varies seasonally.

In a synthesis of the literature, we compared the Oosterschelde with other estuarine-coastal systems focusing on how the spatial phytoplankton gradient is shaped by the distribution of the main environmental drivers. Common spatial phytoplankton patterns include seawards increasing, seawards decreasing, concave with a chlorophyll maximum, weak spatial gradients, and irregular patterns. It should be noted that the spatial phytoplankton pattern is subject to temporal changes and cannot be discussed without specifying the temporal window. For example, the spatial chl-a gradient in this study is different before, during, and after the spring bloom and subject to substantial interannual variability; in river-

dominated systems (Fig. 13), phytoplankton distribution is usually regulated by the episodic, seasonal, and interannual variations of river discharge (Kromkamp and van Engeland, 2010). In addition to natural changes, phytoplankton abundance and spatial heterogeneity can reflect how the lower trophic levels are affected by anthropogenic influences and stressors, such as aquaculture (this study), invasive species (Cloern et al., 2017), and coastal engineering works (Wetsteyn and

Kromkamp, 1994).

**Code and data availability**

The codes for GETM and FABM models are open-access on https://getm.eu/ and https://sourceforge.net/projects/fabm/, respectively. The RWS observational data is accessible on https://www.rijkswaterstaat.nl/water. The NIOZ monitoring data is archived on the NIOZ data repository and available upon

request. The satellite data can be downloaded from the Copernicus Sentinel hub (https://scihub.copernicus.eu).

**Author contributions**

LJ ran the simulations, analyzed the results, and initiated the writing of the manuscript. TG and KS provided guidance and important insights into data interpretation. JCK measured the primary production using the $^{14}CO_2$ uptake experiment. DvdW analyzed the satellite data. JCK and DvdW offered important insight into the phytoplankton dynamics.

PMCDLC and KS built a 1D NPZD model as a basis of the 3D setup. All authors participated in the writing and editing of the manuscript.

**Competing interests.**

No competing interests are present.

**Acknowledgments**

This work is supported by the post-doc framework of Utrecht University and NIOZ, the European Union-funded Horizon 2020 GENIALG (GENetic diversity exploitation for Innovative macro-ALGal biorefinery) project. LJ was also supported by the Fundamental Research Funds for the Central Universities at Hohai University (No. B200201013) and the Natural Science Foundation of Jiangsu Province.

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

**Table 1: Formulations in the biogeochemical model in this study. Parameters and variables in each equation are described in Table 2.**

| | |
|---|---|
| $$T_{fac} = Q_{10}^{(T-T_{ref})/T}$$ | (1) |
| $$\text{DinUpt} = \text{maxUpt} \cdot T_{fac} \cdot \frac{DIN}{DIN + ksDIN} \cdot \frac{PAR}{PAR + ksPAR} \cdot PHY$$ | (2) |
| $$\text{ZooGrz} = \text{maxGrzZoo} \cdot T_{fac} \cdot \frac{PHY}{PHY + ksGrzZoo} \cdot ZOO$$ | (3) |
| $$\text{ZooGro} = (1 - \text{pFaeZoo}) \cdot \text{ZooGrz}$$ | (4) |
| $$\text{ZooExc} = \text{excRZoo} \cdot T_{fac} \cdot ZOO$$ | (5) |
| $$\text{ZooMor} = \text{morRZoo} \cdot T_{fac} \cdot ZOO \cdot ZOO$$ | (6) |
| $$\text{Min} = \text{minR} \cdot T_{fac} \cdot DET$$ | (7) |
| $$\text{BotMin} = \text{minR} \cdot T_{fac} \cdot BDET$$ | (8) |
| $$\text{SinDet} = \text{sinRDet} \cdot DET$$ | (9) |
| $$\text{SinPhy} = \text{sinRPhy} \cdot PHY$$ | (10) |
| $$BivGrz_{i=1,2,3} = maxClr_i \cdot T_{fac} \cdot BIV_i \cdot (PHY + ZOO + DET)$$ | (11) |
| $$BivGro_{i=1,2,3} = (1 - pRspBiv_i) \cdot (1 - pFaeBiv_i - pPsfBiv_i) \cdot BivGrz_i$$ | (12) |
| $$BivExc_{i=1,2,3} = excRBiv_i \cdot T_{fac} \cdot BIV_i$$ | (13) |
| $$\text{MpbDinUpt} = \text{maxUptMpb} \cdot T_{fac} \cdot \frac{DIN}{DIN + ksDINMpb} \cdot \frac{PAR}{PAR + ksPARMpb} \cdot MPB$$ | (14) |
| $$\text{MpbMor} = \text{morRMpb} \cdot T_{fac} \cdot MPB \cdot MPB$$ | (15) |
| $$dDIN/dt = Min + ZooExc - DinUpt + [(1 - pLos)BotMin + BivExc + \sum_{i=1,2,3} pRspBiv_i BivGrz_i - DinUptMpb]/z$$ | (16) |
| $$dPHY/dt = DinUpt - ZooGrz - (SinPhy + \sum_{i=1,2,3} maxClr \cdot T_{fac} \cdot BIV_i \cdot PHY)/z$$ | (17) |
| $$dZOO/dt = ZooGro - ZooExc - ZooMor - \sum_{i=1,2,3} maxClr \cdot T_{fac} \cdot BIV_i \cdot ZOO /z$$ | (18) |
| $$dDET/dt = ZooMor + \text{pFaeZoo} \cdot ZooGrz - Min - (SinDet + \sum_{i=1,2,3} maxClr \cdot T_{fac} \cdot BIV_i \cdot DET)/z$$ | (19) |
| $$dBDET/dt = SinDet + SinPhy - BotMin + \sum_{i=1,2,3} (pFaeBiv_i + pPsfBiv_i) \cdot BivGrz_i + MpbMor$$ | (20) |
| $$dBIV/dt = BivGro - BivExc$$ | (21) |
| $$dMPB/dt = \text{MpbDinUpt} - MpbMor$$ | (22) |

**Table 2: Main variables (bold) and parameters (underlined, followed by values) in equations in Table 1. The parameter values are based on ranges in prior literature (Soetaert et al., 2001; Jiang and Xia, 2017; Wijsman and Smaal, 2017) and tuned for our application.**

| | |
|---|---|
| $T_{fac}$, temperature factor, dimensionless; $T$, *in situ* temperature, °C; $T_{ref}$ = 10 °C, reference temperature; $Q_{10}$ = 2, temperature coefficient | (1) |
| $DIN$, state variable, dissolved inorganic nitrogen, mmol m$^{-3}$; $PHY$, state variable, phytoplankton biomass, mmol m$^{-3}$; $DinUpt$, pelagic DIN uptake, mmol m$^{-3}$ d$^{-1}$; $PAR$, *in situ* photosynthetically active radiation, µmol-photons m$^{-2}$ s$^{-1}$; $maxUpt$ = 1.7 d$^{-1}$, maximum DIN uptake rate; $ksDIN$ = 1 mmol m$^{-3}$, half-saturation DIN concentration; $ksPAR$ = 140 µmol-photons m$^{-2}$ s$^{-1}$, half-saturation PAR | (2) |
| $ZOO$, state variable, zooplankton biomass, mmol m$^{-3}$; $ZooGrz$, zooplankton grazing, mmol m$^{-3}$ d$^{-1}$; $maxGrzZoo$ = 0.8 d$^{-1}$, maximum zooplankton grazing rate; $ksGrzZoo$ = 0.6 mmol m$^{-3}$, half-saturation phytoplankton concentration for zooplankton grazing | (3) |
| $ZooGro$, zooplankton growth, mmol m$^{-3}$ d$^{-1}$; $pFecZoo$ = 0.3, fraction of zooplankton faeces in total grazing | (4) |
| $ZooExc$, zooplankton excretion, mmol m$^{-3}$ d$^{-1}$; $excRZoo$ = 0.08 d$^{-1}$, zooplankton excretion rate | (5) |
| $ZooMor$, zooplankton mortality, mmol m$^{-3}$ d$^{-1}$; $morRZoo$ = 0.45 m$^3$ mmol$^{-1}$ d$^{-1}$, quadratic zooplankton mortality rate | (6) |
| $DET$, state variable, pelagic detritus, mmol m$^{-3}$; $Min$, DIN regeneration from pelagic detritus, mmol m$^{-3}$ d$^{-1}$; $minR$ = 0.02 d$^{-1}$, mineralization rate | (7) |
| $BDET$, state variable, benthic detritus, mmol m$^{-2}$; $BotMin$, DIN regeneration from benthic detritus, mmol m$^{-2}$ d$^{-1}$ | (8) |
| $SinDet$, detritus sinking, mmol m$^{-2}$ d$^{-1}$; $sinRDet$ = 1.0 m d$^{-1}$, sinking rate of detritus | (9) |
| $SinPhy$, phytoplankton sinking, mmol m$^{-2}$ d$^{-1}$; $sinRPhy$ = 0.2 m d$^{-1}$, sinking rate of phytoplankton | (10) |
| $BIV_{1,2,3}$, state variable, biomass of bivalve filter feeding mussels, oysters, and cockles (1–3 denotes these three species hereafter), mmol m$^{-2}$, $BivGrz_{1,2,3}$, bivalve grazing rate, mmol m$^{-2}$ d$^{-1}$; $maxClr_{1,2,3}$ = 0.007, 0.015, 0.0037 m$^3$ mmol$^{-1}$ d$^{-1}$ | (11) |
| $BivGro_{1,2,3}$, bivalve growth, mmol m$^{-2}$ d$^{-1}$; $pRspBiv_{1,2,3}$ = 0.001, 0.003, 0.001, bivalve respiration portion in the net assimilation; $pFecBiv_{1,2,3}$ + $pPsfBiv_{1,2,3}$ = 0.55, 0.39, 0.33, fraction of bivalve faeces (*Fec*) and pseudo-faeces (*Psf*) in total grazing | (12) |
| $BivExc_{1,2,3}$, bivalve excretion, mmol m$^{-2}$ d$^{-1}$; $excRBiv_{1,2,3}$ = 0.0006, 0.0001, 0.001 d$^{-1}$, bivalve excretion rates | (13) |
| $MPB$, state variable, microphytobenthos biomass, mmol m$^{-2}$; $DinUptMpb$, microphytobenthic DIN uptake, mmol m$^{-2}$ d$^{-1}$; $maxUpt$ = 0.75 d$^{-1}$, maximum microphytobenthic DIN uptake rate; $ksDINMpb$ = 1 mmol m$^{-3}$, half-saturation DIN concentration for microphytobenthos; $ksPARMpb$ = 100 µmol-photons m$^{-2}$ s$^{-1}$, half-saturation PAR for microphytobenthos | (14) |
| $MpbMor$, microphytobenthos mortality, mmol m$^{-2}$ d$^{-1}$; $morRMpb$ = 0.001 m$^2$ mmol$^{-1}$ d$^{-1}$, quadratic microphytobenthos mortality rate | (15) |
| $dDIN/dt$, $dPHY/dt$, $dZOO/dt$, $dDET/dt$, change rates of pelagic variables, mmol m$^{-3}$ d$^{-1}$; $dBDET/dt$, $dBIV/dt$, $dMPB/dt$, change rates of benthic variables, mmol m$^{-2}$ d$^{-1}$; $z$, thickness of the bottom layer, m | (16–22) |

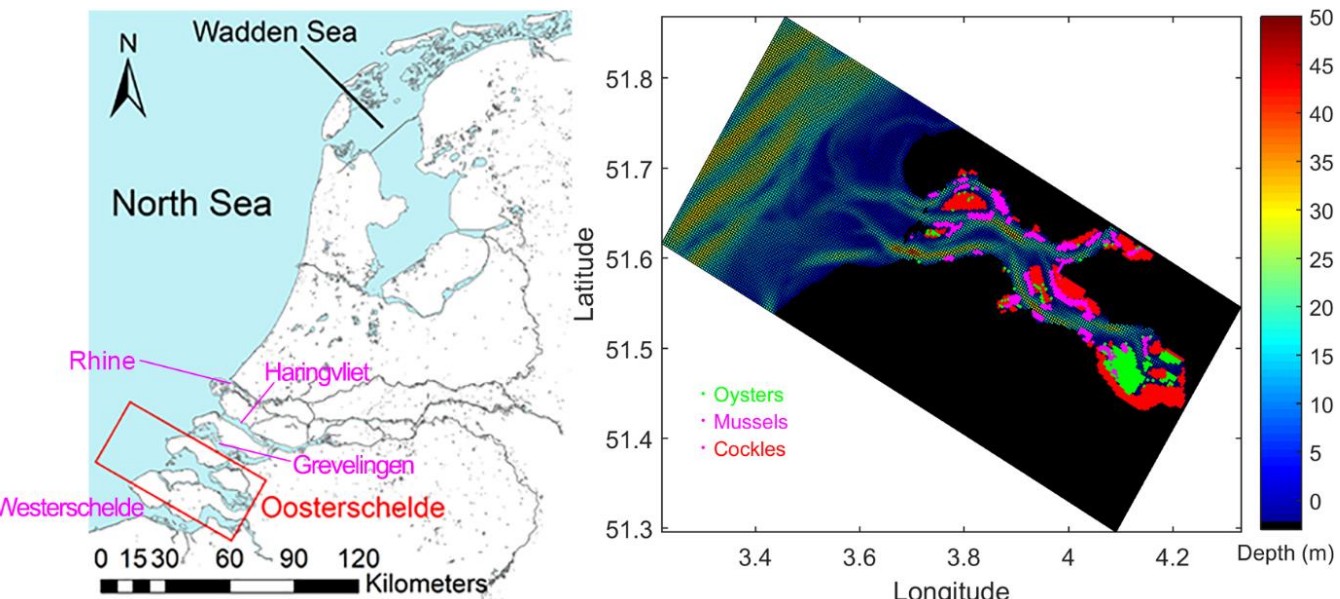

**Figure 1: Geographic location of the Oosterschelde and the GETM-FABM model grid, domain, and bathymetry. Green, pink, and red dots in the right panel indicate the distribution of three dominant bivalve species in the Oosterschelde, oysters, mussels, and cockles (data source: Wageningen Marine Research).**

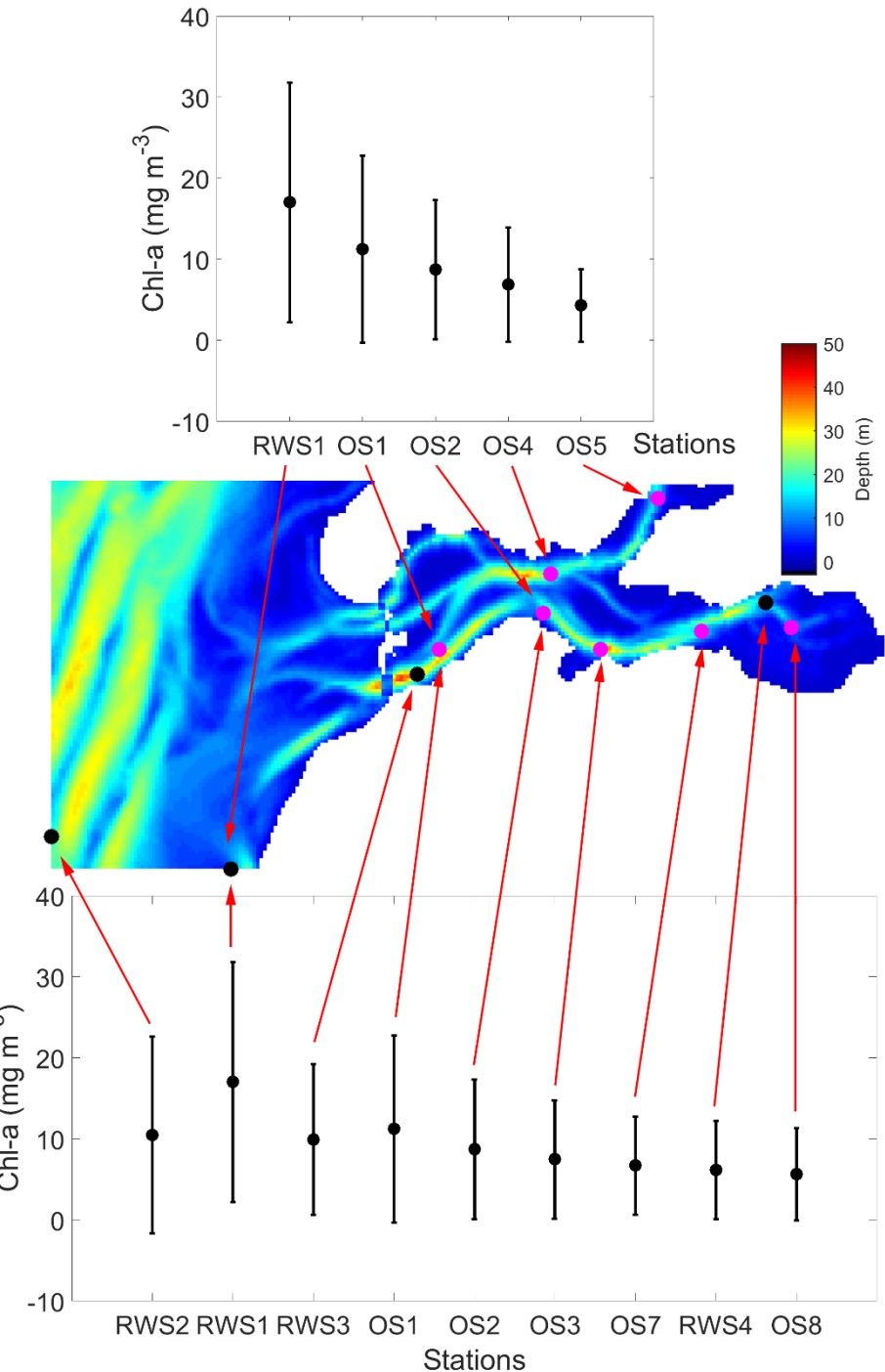

**Figure 2: Average spring (March to May) chlorophyll a (chl-a) concentration during 1995–2013 at NIOZ (OS1–OS8) and Rijkswaterstaat (RWS1–RWS4) monitoring stations. The error bars of each stations indicate standard deviations. The map shows bathymetry in the GETM-FABM model domain denoted in Figure 1 and marks all NIOZ and RWS stations within the domain. RWS1–RWS4 in this study are short names for Walcheren 2 km, Walcheren 20 km, Wissenkerke, and Lodijkse Gat in the RWS database, respectively.**

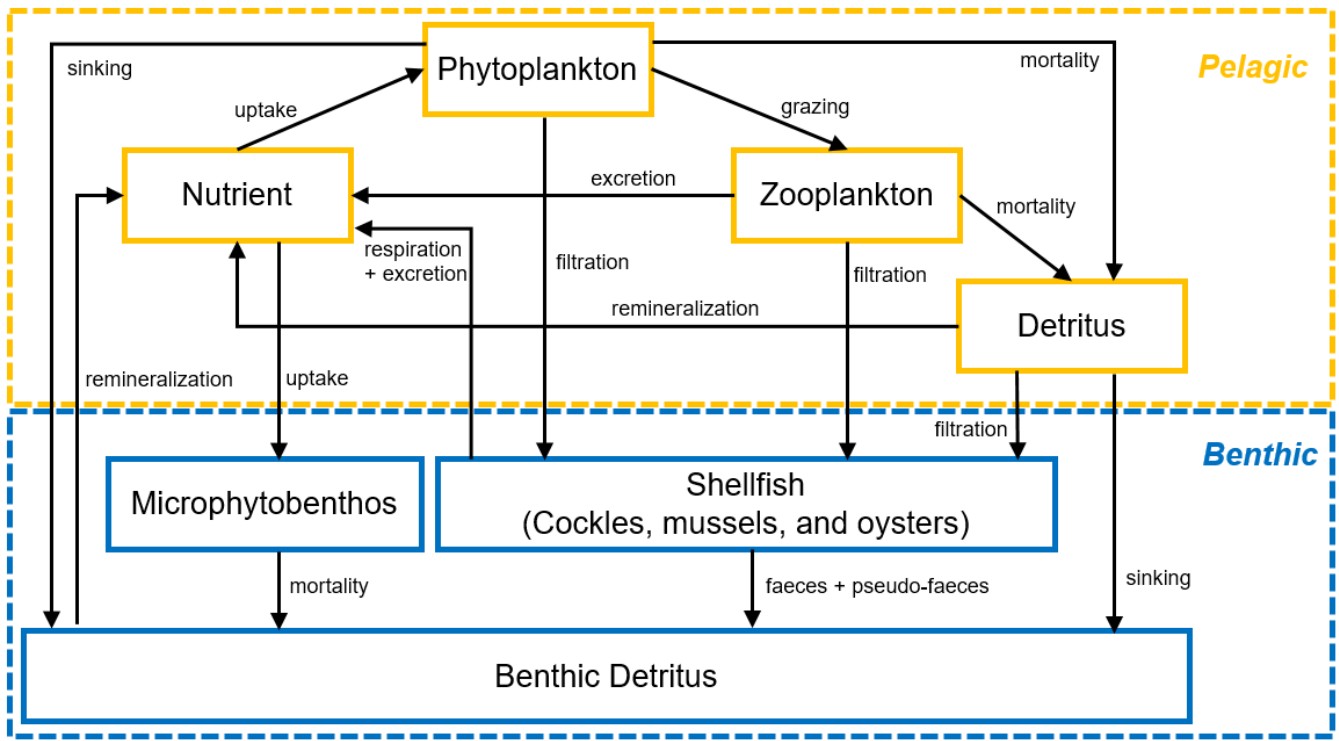

**Figure 3: Conceptual diagram of the nitrogen-based seven-variable biogeochemical model structure in FABM. Box and arrows denote state variables and fluxes of nitrogen, respectively.**

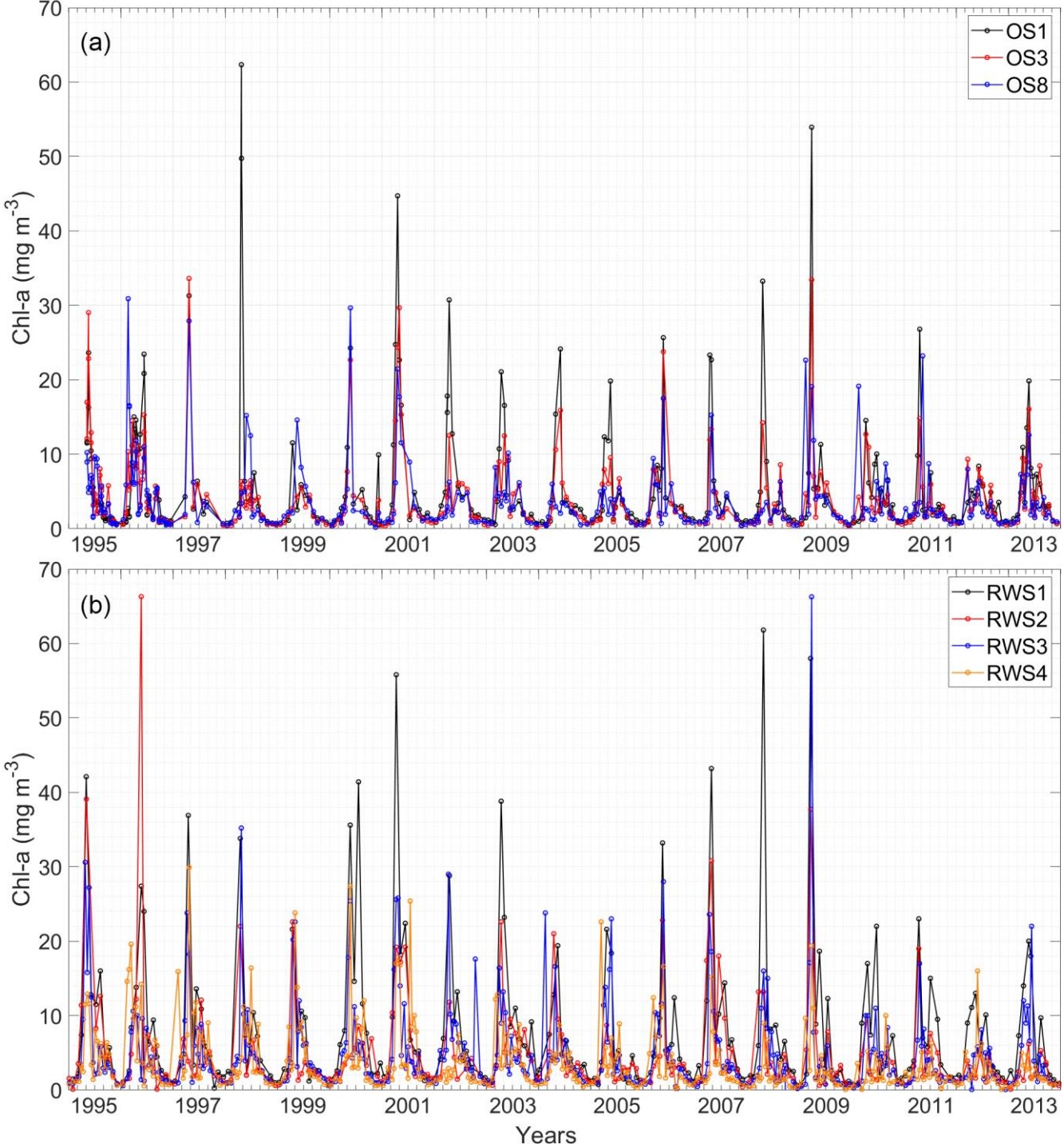

**Figure 4: Time series of chlorophyll a (chl-a) concentrations during 1995–2013 at (a) NIOZ stations OS1, OS3, and OS8 and (b) Rijkswaterstaat stations RWS1–RWS4. Intervals between grid lines indicate two months. See Figure 2 for station locations.**

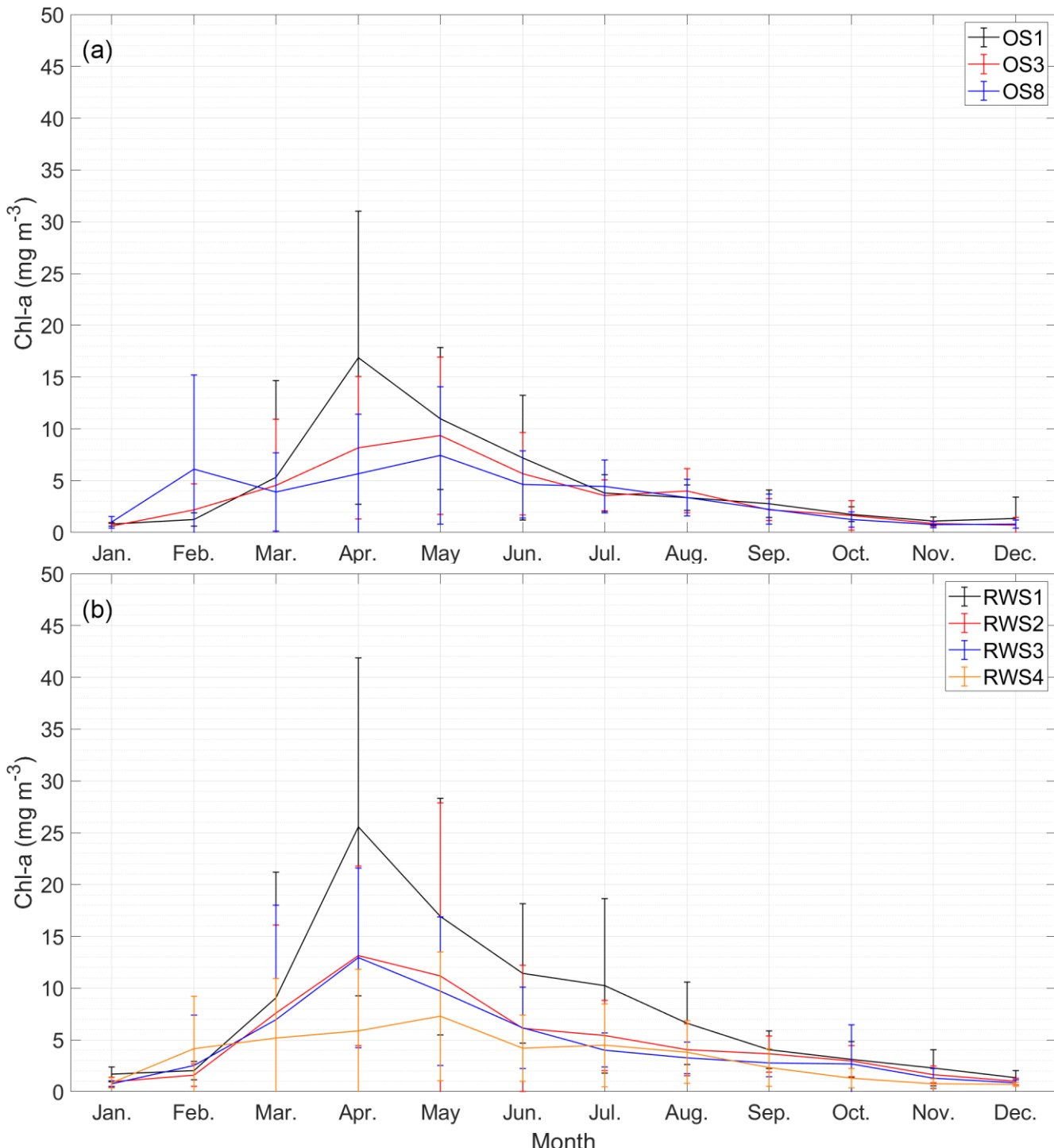

**Figure 5: Monthly average chlorophyll a (chl-a) concentrations during 1995–2013 at (a) NIOZ stations OS1, OS3, and OS8 and (b) Rijkswaterstaat stations RWS1–RWS4. The error bars of each station indicate standard deviations. See Figure 2 for station locations.**

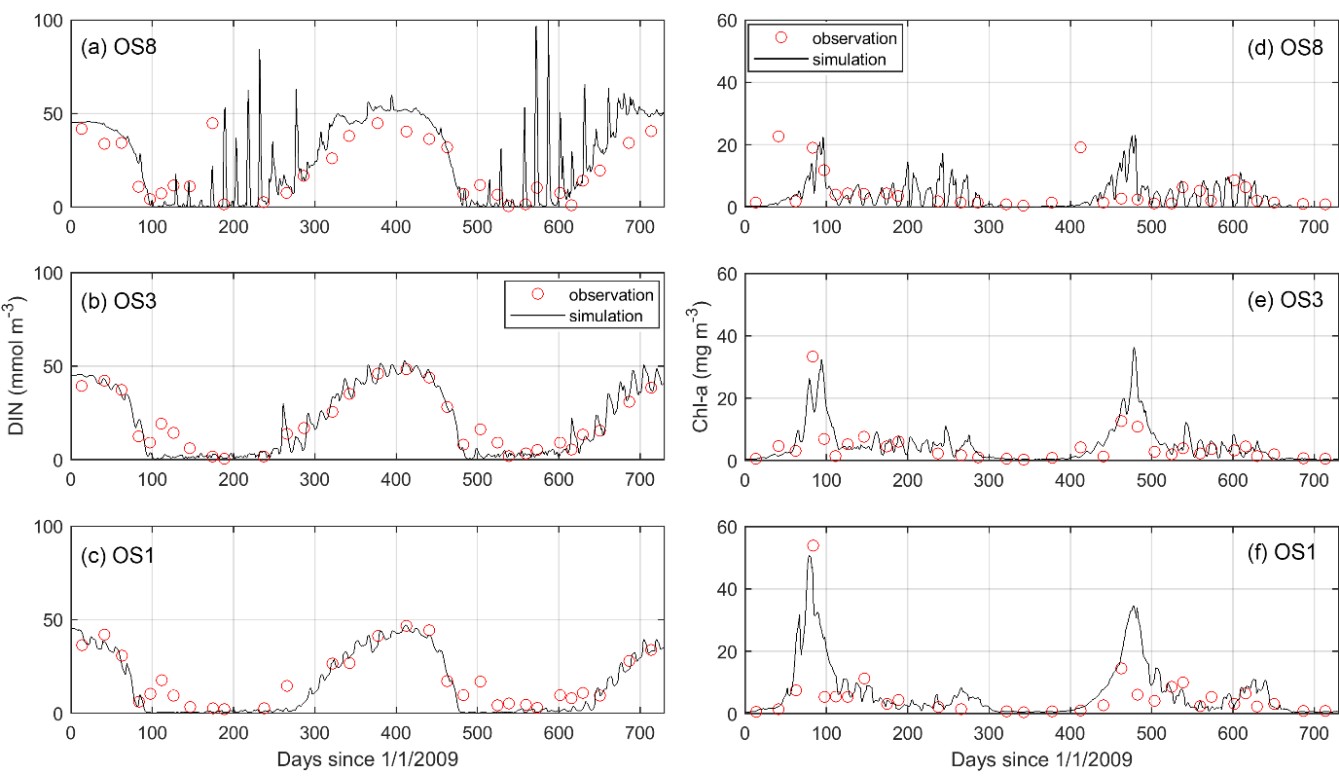

**Figure 6: Comparison between simulated and observed dissolved inorganic nitrogen (DIN) and chlorophyll a (chl-a) in the years 2009–2010 at stations OS8, OS3, and OS1. See Figure 2 for station locations.**

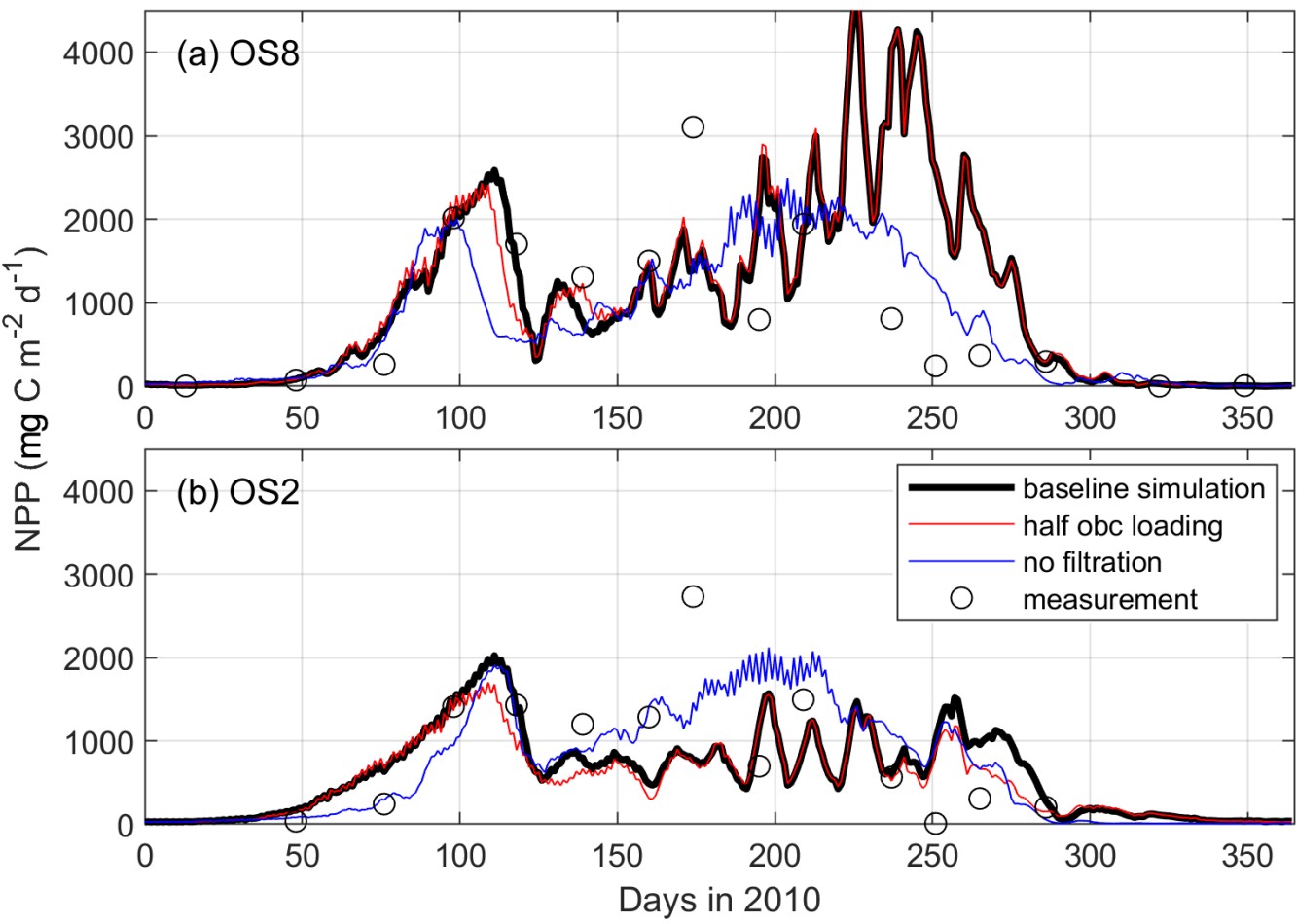

**Figure 7: Comparison between modeled and measured depth-integrated net primary production (*NPP*, mg carbon m$^{-2}$ d$^{-1}$) in 2010 at stations OS8 and OS2. The three model scenarios include the baseline scenario, halving the open boundary nutrient and phytoplankton loading, and switching off bivalve filtration feeders.**

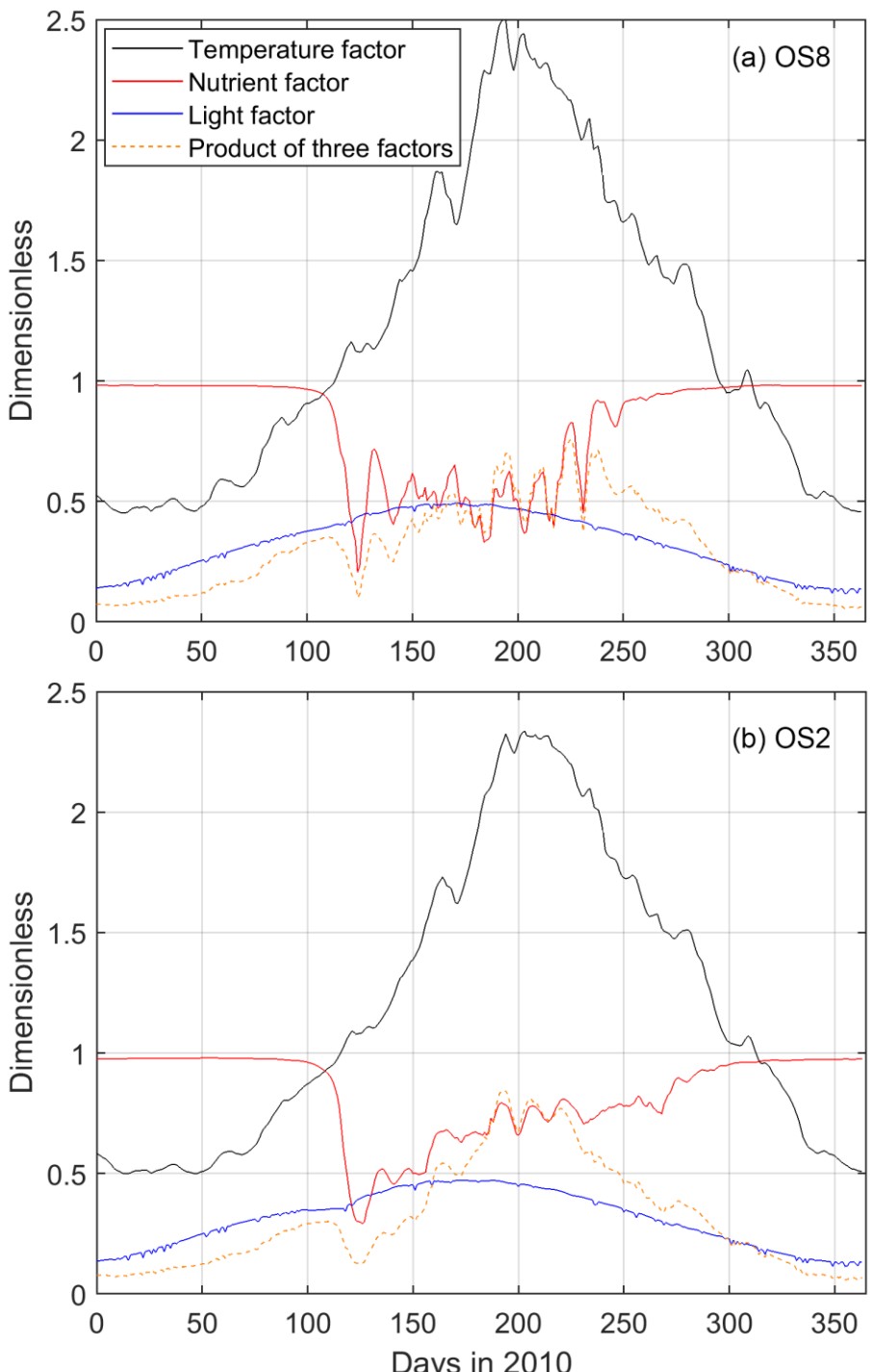

**Figure 8: Modeled surface temperature, nutrient, and light factors affecting the phytoplankton growth rate at (a) OS 8 and (b) OS2. Temperature factor is calculated as Equation (1) in Table 1. Nutrient factor = *DIN* / (*DIN* + *ksDIN*), light factor = *PAR* / (*PAR* + *ksPAR*), as shown in Equation (2) in Table 1. According to Equation (2), the product of these three factors is positively related to phytoplankton growth rate.**

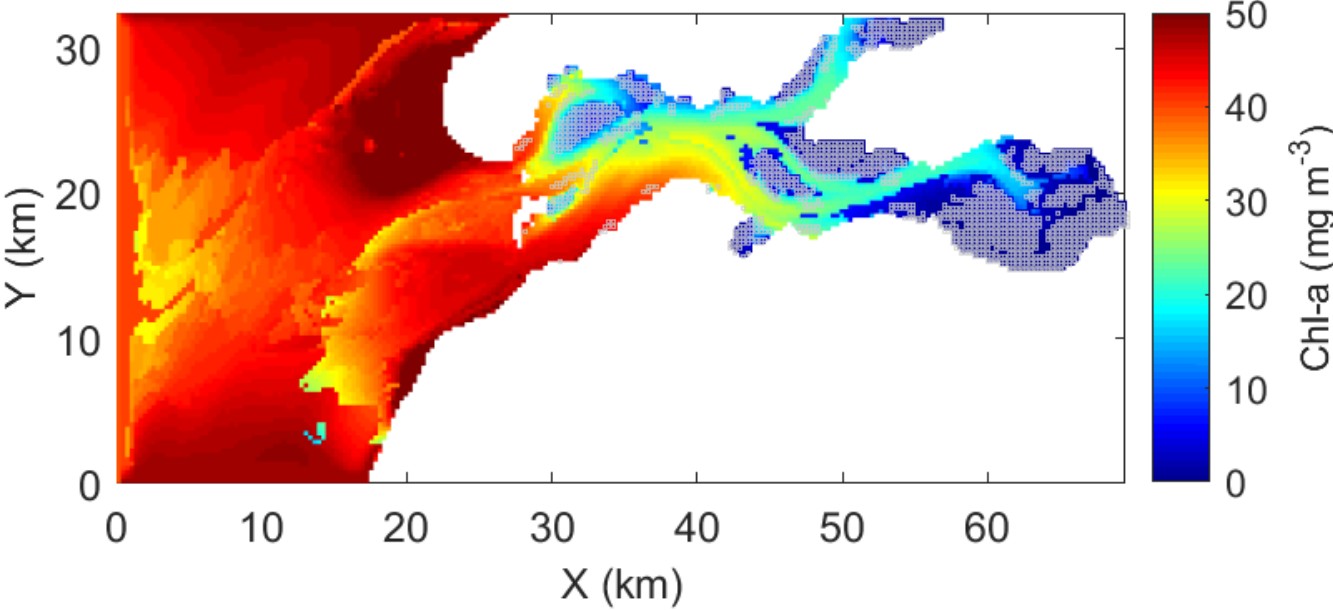

**Figure 9: The 15-day (5–19 March) average of modeled chlorophyll a (chl-a) during the peak spring bloom in 2009. Grey squares indicate the locations of wild and cultured shellfish as in Figure 1.**

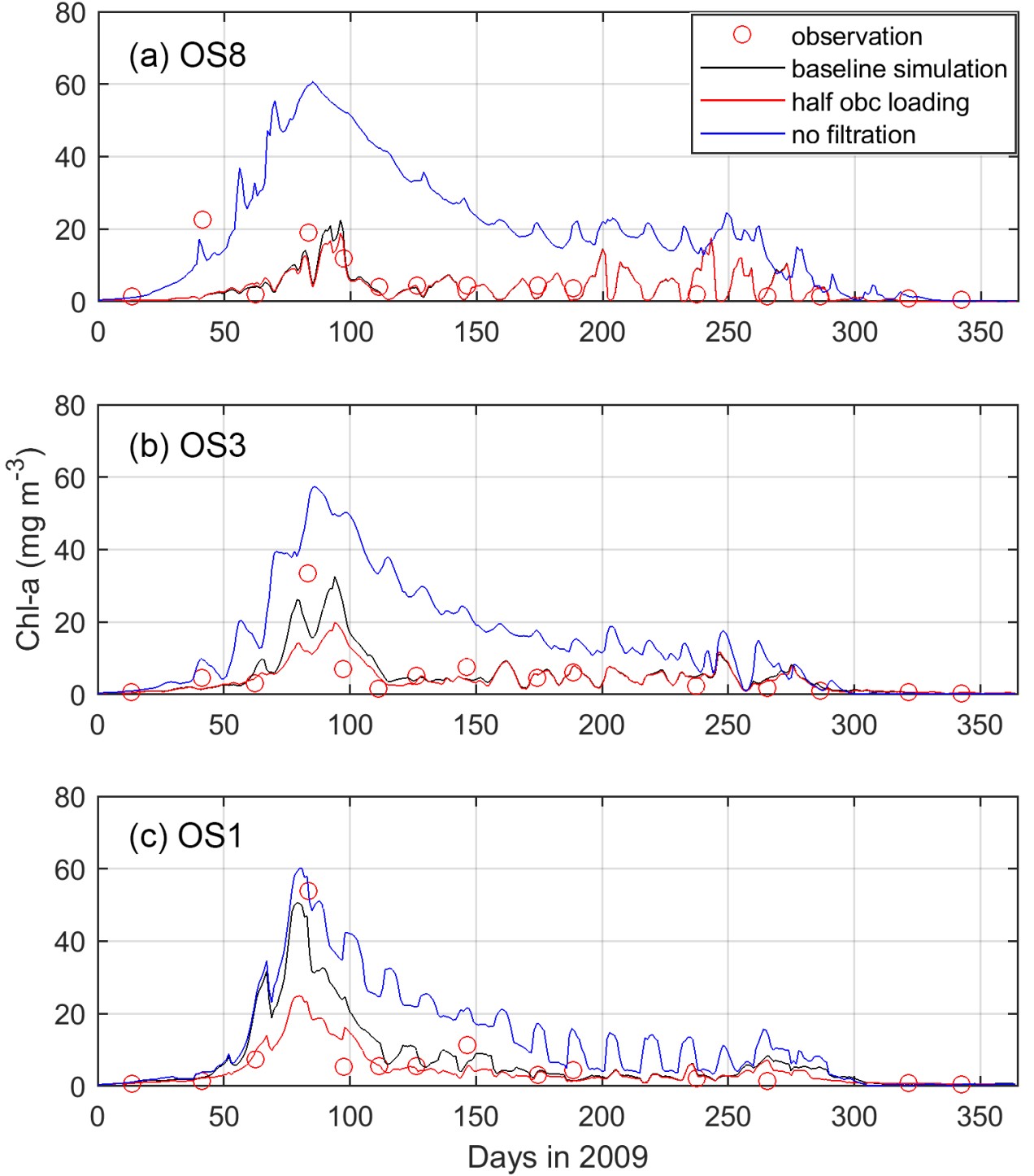

**Figure 10: Modeled and observed chlorophyll a (chl-a) in 2009 at stations OS8, OS3, and OS1. The three model scenarios include the baseline scenario, halving the open boundary nutrient and phytoplankton loading, and switching off bivalve filtration feeders. See Figure 2 for station locations.**

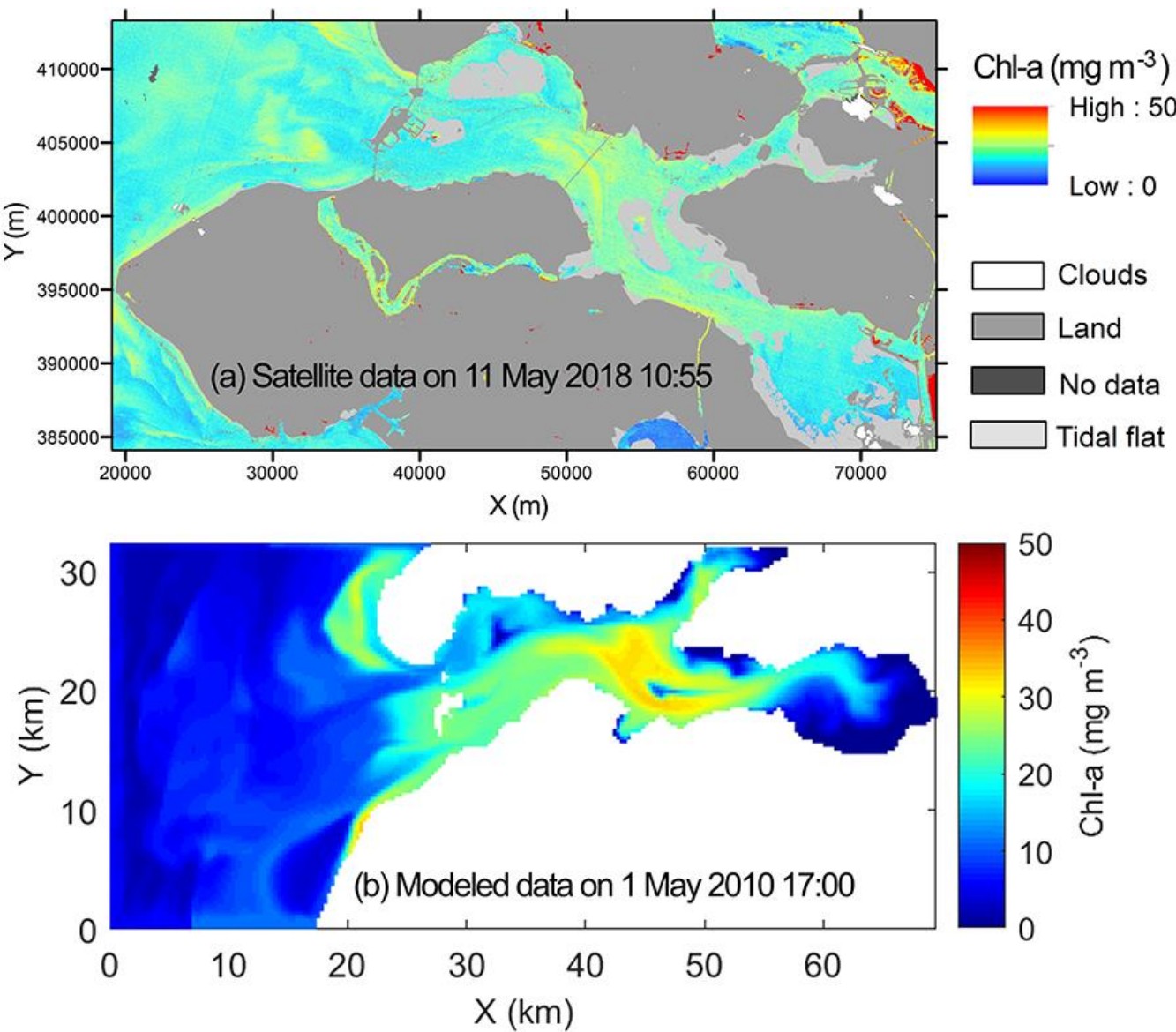

**Figure 11. Chlorophyll a (chl-a) in the Oosterschelde retrieved from (a) a high tide Sentinel-2 MSI image of 11 May 2018, at 10:55 masking tidal flats from a low tide Sentinel-2 MSI image of 21 April 2019 and (b) the model on 1 May 2010, at 17:00. Both snapshots are during high tide. The coordinate system in (a) is Amersfoort / RD New.**

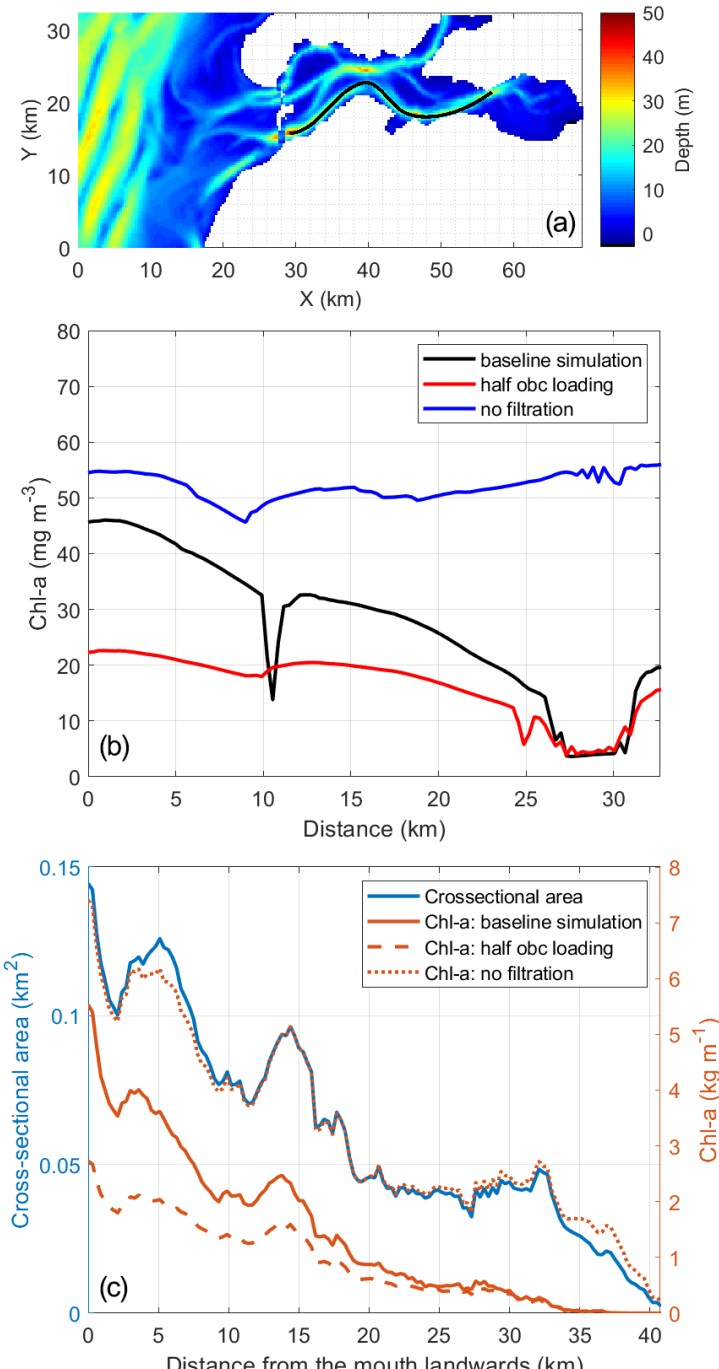

**Figure 12. The 15-day (5–19 March) average of modeled chlorophyll a (chl-a) during the peak spring bloom in 2009 (b) along a transect over the southern channel of the Oosterschelde and (c) cross-sectionally integrated from the bay mouth landwards. The transect location is shown in (a). The blue line in (c) denotes the cross-sectional area used for integration. The northern branch (a) is excluded from the calculation in (c) because of a different orientation of channels. The distance on the x-axis of panels (b) and (c) is from west to east. The three model scenarios include the baseline scenario, halving the open boundary nutrient and phytoplankton loading, and switching off bivalve filtration feeders.**

| Common spatial gradients | Example ecosystems and references | Flushing mechanisms | Main drivers of phytoplankton biomass |
|---|---|---|---|
| Seawards increasing (Phytoplankton biomass vs Land–Ocean) | (1) Oosterschelde, the Netherlands (this study) | Tide-dominated | Grazing loss and tidal import |
| | (2) Rías Baixas of Galicia, Spain (Figueiras et al., 2002); Willapa Bay, USA (Hickey and Banas, 2003; Banas et al., 2007) | Tide-dominated | Tidal import |
| | (3) Chilika Lagoon, India (Srichandan et al., 2015) | River-dominated | Light limitation |
| Seawards decreasing (Phytoplankton biomass vs Land–Ocean) | (1) Westerschelde estuary, the Netherlands and Belgium (Kromkamp and Peene, 1995; Krompkamp et al., 1995; Muylaert et al., 2005; Soetaert et al., 1994, 2006) | River and tides, or river-dominated | Salinity stress, grazing loss, and transport |
| | (2) Chesapeake Bay outflow plume, USA (Jiang and Xia, 2018); Mississippi River plume, USA (Gomez et al., 2018) | River and tides | Nutrient limitation |
| Chl-a maximum (Phytoplankton biomass vs Land–Ocean) | (1) Chesapeake Bay, USA (Jiang and Xia, 2017); Delaware Bay, USA (Fisher et al., 1988); York River, USA (Sin et al., 1999); Neuse-Pamlico estuary, USA (Valdes-Weaver et al., 2006); Logan River and Moreton Bay, Ausatralia (O'Donohue and Dennison, 1997) | River and tides, or river-dominated | Upper reach limited by light or transport loss; lower reach limited by nutrients |
| Spatially uniform or weak spatial gradient (Phytoplankton biomass vs Land–Ocean) | (1) San Francisco Bay, USA , after 1987 (Cloern et al., 2017; Kimmerer and Thompson, 2014) | River and tides | Grazing loss |
| | (2) Hudson River estuary, USA (Fisher et al., 1988; Howarth et al., 2000; Strayer et al., 2008) | River-dominated | Transport and grazing loss |
| Patches/irregular patterns (Phytoplankton biomass vs Land–Ocean) | (1) Baie des Veys estuary, France (Grangeré et al., 2010) | River and tides | Grazing loss |
| | (2) Krka estuary, Croatia (Ahel et al., 1996) | River-dominated | Point-source nutrient input |
| | (3) St. Lucia estuary, South Africa (van der Molen and Perissinotto, 2011) | River-dominated | DIN:DIP ratio, salinity, temperature, and irradiance |

**Figure 13. Common spatial patterns of phytoplankton biomass in estuarine-coastal systems. For comparison with the Oosterschelde, example ecosystems for each type are given along with references, the dominant flushing mechanisms, and main drivers of phytoplankton accumulation.**