# Peer review of "Drivers of the spatial phytoplankton gradient in estuarine-coastal systems: generic implications of a case study in a Dutch tidal bay"

_Biogeosciences, 2020_

## Referee Comment (RC1) · Tom Fisher (Referee) · 6 Mar 2020

This is a well-written, multi-disciplinary manuscript addressing the distribution of phytoplankton and primary productivity in a Dutch estuary. The three approaches (observational, modeling, remote sensing) provide a strong basis for describing phytoplankton distributions and the causes of the patterns. The literature synthesis at the end of the manuscript sets the results of this manuscript in global context. Overall, a strong addition to the estuarine literature.

I suggest that the authors address a few issues that I think are missing: 1.Light limitation of phytoplankton growth is common in estuaries and often occurs in turbid, nutrientrich, low-salinity waters with no vertical stratification. No data are presented on salinity or density in this manuscript, and the authors simply state that there is no stratification, citing another paper. They could be right, and water depth may limit vertical mixing. However, my experience suggests that vertical stratification in spring under high river flow conditions initiates the spring bloom. Even if there are no CTD data available to calculate vertical variations in density, the authors should at least mention the possibility that the low salinity areas with high nutrients and turbidity may be light-limited regions of the estuary.

2.I am surprised that there are no data presented on salinity, temperature, river discharge, and river nutrient concentrations. The authors nicely show that advective inputs of shelf nutrients and phytoplankton is likely to be small, but never explore the role of river inputs. This could be a whole other paper, but they could at least mention that riverine inputs, both freshwater and nutrients, are likely driving the spring bloom. They could do additional model runs with half of the river discharge or half of the river N concentrations, but this might be more work than reasonable. I suspect that vertical stratification in the inner half of the estuary allows algal biomass to accumulate following high winter-spring river discharge. What would the model show if freshwater flow and/or river nutrient concentrations were halved? Can any of the temporal variations in the spring bloom be related to river flow?

3. I made a few minor grammatical or wording suggestions to the pdf of the text and for improvements in the figures that will be easy to address. This isn't a long manuscript, and the above two issues can probably be addressed briefly in 1-2 pages.

Please also note the supplement to this comment: https://www.biogeosciences-discuss.net/bg-2020-40/bg-2020-40-RC1-supplement.pdf

---

## Referee Comment (RC3) · Nicole Millette (Referee) · 11 Mar 2020

This is one of the most well-written papers I have reviewed and everything was well laid out and easy to follow. However, I was left with wanting a lot more information and discussion from the authors. This is the first time I have ever said this, but this paper is too short. There are three different approaches used in the paper, which is a plus, but there is a lot of information that could be shared about each approach. The main conclusions for the paper are clear and well supported, but this is a case study, and I think there should be a lot more acknowledgement and discussion of the nuances in the variability of the spring bloom in the Oosterschelde; it has not always followed the

described pattern. More specifics are provided below:

General Comments 1. More information and discussion of the field data (a) Page 6, line 17-20: There is some discussion in here about how nutrients, light, and temperature effect the phytoplankton biomass annual cycle, but none of the data presented demonstrate or support these claims. What is this based off of, other people's findings or the authors own analysis? If these are the authors own conclusions, then I would like to see data and analysis to support these claims. (b) A figure of DIN concentrations, similar to figure 4, would be beneficial. (c) Why is OS6 not included in figure 2? (d) Page 4, line 16-17: More information on the 14CO2 uptake experiments would be helpful - Who did the experiments? At all stations? For all sample dates in 2010? Is this data published elsewhere? (e) The SD bars in figure 2 are large for both gradients, suggesting a wide range of [chla] at all stations during spring between 1995 and 2013. The average values show a tendency towards higher [chla] at the mouth, but the large SD demonstrate a lot of inter-annual variability. Based on figure 4a, it appears that the pattern in spring phytoplankton biomass described in this paper dominated between 2000 and 2009. Pre-2000, [chla] at the head and mid-bay repeatedly matched or surpassed the mouth, 1998 being the clear exception. After 2009, [chla] during spring appears to become less distinct between each location. This does not negate the conclusions of the paper, but the years that do not match the pattern should be acknowledged. It is not expected that every year will always be the same, so what might have happened in the years that didn't follow the pattern?

2. More information and discussion of the model results (a) Figure 6, 7, 9: Are the field observations in these figures averages? If there is any way to calculate the standard deviation for these values, it should be included. (b) Page 7, line 12: The authors mention that before the bloom, phytoplankton biomass and growth rates were low, but this data is not really presented. I know [chla] = phytoplankton biomass, so maybe keep the terminology consistent in the paper. However, the growth rate data from the model is something that I think should be presented, it sounds like it is interesting. (c) Page

7, line 17-18: I wanted more detail here, rather than saying NPP is generally higher at OS8 compared to OS2. The authors note that the model overestimates NPP at OS8 in the fall 2010, but that overestimation makes it difficult to compare the two sites in figure 7. What is the average + SD observed NPP at each station? Is it significantly different? On average, how much higher is NPP at OS8 compared to OS2 in observed and modeled data? (d) Figure 7: The authors explain the overestimation at OS8, but why did the model miss the highest peak in NPP at both stations around day 175?

3. My personal opinion is the synthesis section (6) does not belong in this paper. Figure 12 and all the work the authors did is very interesting, but does not fit with the rest of the paper. Figure 12 and the synthesis sections should be its own paper. There is a lot of information in figure 12 that deserves more than three paragraphs of explanation. A case study of chlorophyll a spatial pattern in the Oosterschelde is not the place to propose a categorization of chlorophyll a spatial patterns for all estuaries.

Specific Comments 1. Introduction Page 2, line 28 – Page 3, line 1: Include the pros and cons of all three methods. No cons are mentioned for ecological methods.

2. Methods Page 5, line 24-26: The model output results went through two conversions to match the observation data (N->C, then C->chla). There should be some mention of the assumptions and limitations of these conversions because they are not perfect and there has recently been growing criticism of the C:chla ratio.

7. Summary Page 11, line 24-28: These last two sentences do not accurately sum up the paper. There is no discussion of temporal variability in the spatial distribution of phytoplankton in Oosterschelde and I did not get an understanding of phytoplankton's role as an ecological indicator. The paper would be greatly strengthened by a discussion of the temporal variability and what caused it.

---

## Referee Comment (RC4) · J. Blake Clark (Referee) · 7 Apr 2020

General Comments This paper describes a coupled observational, modeling and satellite observational study of an estuarine system in the North Sea. Overall, the story and results were well conveyed and the conclusions regarding drivers of spatial and temporal variability in the estuary were supported. The main take away is that there is a Type I phytoplankton distribution and it is mainly driven by benthic grazing pressure in the landward stations. The model supports the importance of grazing pressure on the spatial distribution by numerically removing bivalves in the modeling system. Modeling estuarine primary production and chl-a distribution can be particularly challenging,

and I think the author's did a pretty good job at capturing overall NPP magnitude and some of the temporal variability, compared to 14-C NPP incubation data. The synthesis at the end is particularly useful, especially related to the discussion of how different mechanisms can lead to similar patterns of chl-a distribution, depending on the system.

The main methodology and results that need to be improved upon, or omitted, relates to the use of the satellite observations. The author's use one image (Fig. 10) and it doesn't really track with the results and conclusions of the rest of the paper. In fact, the chl-a concentration is highest in the landward stations where in most observations showed lower chl-a concentration. I understand the desire to do this coupled methodological approach, but in my opinion if satellite data is to be used, it should be developed a bit more to support the observational and modeling work. There is definitely a lot of value in using these data, but acquiring more spring bloom images from MERIS data that fall within the observational window would offer a bit more support for the other results.

Specific Comments Page 2 Line 10: This sentence with the semi-colons is oddly structured, consider revising because the information is good.

4-20: "Light attenuation was measured . . ." How specifically was light attenuation measured and with what instrument?

4-25: "We used the measured values . . ." I don't quite understand this sentence, consider revising

5-510: What weather forcing was used, specifically, and how was surface irradiance specified?

5-15: I see in the equations, detritus sinking is also calculated, but perhaps mention that here as well

5-30: From what I can tell the bivalve biomass is constant, but perhaps clarify that here. Are the bivalves growing and dying or are they constant in time?

6-20:25: It would be useful to show some kind of climatology of the measurements with a window or errorbars that show the inter-annual variability. See figs in Testa, J. M., Murphy, R. R., & Brady, D. C. (2018). Nutrient-and climate-induced shifts in the phenology of linked biogeochemical cycles in a temperate estuary. Frontiers in Marine. https://www.frontiersin.org/articles/10.3389/fmars.2018.00114

9-20: Does the decreasing depth (presumably) also cause the benthic-pelagic coupling to become stronger? Are there bivalves in the more seaward stations but because there is a greater volume of water the grazing pressure just is less, on an areal basis?

---

## Author Comment (AC3) · 11 May 2020

**General Comments**

This paper describes a coupled observational, modeling and satellite observational study of an estuarine system in the North Sea. Overall, the story and results were well conveyed and the conclusions regarding drivers of spatial and temporal variability in the estuary were supported. The main take away is that there is a Type I phytoplankton distribution and it is mainly driven by benthic grazing pressure in the landward stations. The model supports the importance of grazing pressure on the spatial distribution by numerically removing bivalves in the modeling system. Modeling estuarine primary production and chl-a distribution can be particularly challenging, and I think the author's did a pretty good job at capturing overall NPP magnitude and some of the temporal variability, compared to 14-C NPP incubation data. The synthesis at the end is particularly useful, especially related to the discussion of how different mechanisms can lead to similar patterns of chl-a distribution, depending on the system.

**Response (1): Thanks for the positive feedback and the following suggestions. We have revised the manuscript as suggested and replied to the comments point by point.**

The main methodology and results that need to be improved upon, or omitted, relates to the use of the satellite observations. The author's use one image (Fig. 10) and it doesn't really track with the results and conclusions of the rest of the paper. In fact, the chl-a concentration is highest in the landward stations where in most observations showed lower chl-a concentration. I understand the desire to do this coupled methodological approach, but in my opinion if satellite data is to be used, it should be developed a bit more to support the observational and modeling work. There is definitely a lot of value in using these data, but acquiring more spring bloom images from MERIS data that fall within the observational window would offer a bit more support for the other results.

**Response (2): Thank you for the suggestions.**

The Envisat MERIS satellite data can be used to retrieve chl-a concentrations at a spatial concentration of ca 300 m (FR, full resolution data) to ca 1 km (RR, reduced resolution data). Their spatial resolutions are typically not sufficiently high for the application of the Oosterschelde, because the narrowest portions of the basin and the northern branch are around 3 km, and the Oosterschelde has intertidal flats that fall

dry during low water (Fig. R1). Therefore, the land or intertidal pixels may interfere with the MERIS data in the Oosterschelde region.

van der Woerd et al. (2011) have investigated the surface chl-a in the North Sea with MERIS reduced resolution output. Their processed MERIS satellite data show that results within the Oosterschelde should be treated with caution, especially at narrow regions (Fig. R2). In contrast, the 10-m-resolution Sentinel-2 MSI data applied in our study have a spatial resolution suitable for the Oosterschelde.

In addition to the demand of a high spatial resolution, the satellite data to be used in our study also needs to be taken at high tides. One third area of the Oosterschelde is covered by intertidal flats and the water around the flats is extremely shallow (Nienhuis and Smaal, 1994). Bottom reflectance may become another source of errors at low tides (cf Arabi et al., 2019, for optically shallow water effects from MERIS full resolution images of the Wadden Sea). Availability of high tide, low cloud cover images that can show the overall spatial chl-a gradient in the Oosterschelde. are hence very limited.

Moreover, the satellite image in Fig. 11a offers, as a snapshot, valuable insight into the spatial gradient. The spatial phytoplankton pattern shown in Fig. 11a was also detected in the model output (Fig. 11b), which partly validates the model. Although the seaward increasing chl-a gradient is most common in the Oosterschelde in spring, it changes with time. That is, when discussing the spatial phytoplankton variability, we have to be aware of the temporal variability. When describing the general spatial gradient, the less general "exceptions" needs to be noticed. In the revised manuscript, we have emphasized the importance of temporal variability. The satellite image and the less frequent spatial gradient it displays (Fig. 11) fits in that standpoint. Thereby, we tend to retain Section 4.3 and Fig. 11a.

Figure R1: The cross-sectional area and width of the Oosterschelde from the mouth to its eastern end. The northern branch (Fig. 1) is excluded from the calculation because of a different orientation of channels. This figure is Fig. 6 in Jiang et al., 2020.

**References**

- Arabi, B., Salama, M. S., Van der Wal, D., Pitarch, J., and Verhoef, W.: The impact of sea bottom effects on the retrieval of water constituent concentrations from MERIS and OLCI images in shallow tidal waters supported by radiative transfer modeling, Remote Sens. Environ., 237, 11596, https://doi.org/10.1016/j.rse.2019.111596, 2020.
- Jiang, L., Gerkema, T., Idier, D., Slangen, A. B. A., and Soetaert, K. E.: Effects of sea-level rise on tides and sediment dynamics in a Dutch tidal bay, Ocean Sci., 16, 307–321, https://doi.org/10.5194/os-16-307-2020, 2020.
- Nienhuis, P. H., and Smaal, A. C.: The Oosterschelde estuary, a case-study of a changing ecosystem: an introduction, Hydrobiologia, 282/283, 1–14, http://doi.org/10.1007/BF00024616, 1994.
- van der Woerd, H. J., Blauw, A., Peperzak, L., Pasterkamp, R., and Peters, S.: Analysis of the spatial evolution of the 2003 algal bloom in the Voordelta (North Sea), J. Sea Res., 65, 195–204, ttp://doi.org/10.1016/j.seares.2010.09.007, 2011.

Specific Comments

Page 2 Line 10: This sentence with the semi-colons is oddly structured, consider revising because the information is good.

Response (3): This sentence is rephrased and split into three sentences. Please see Page 2 Lines 10–13 in the "accept-changes" version of the revised manuscript.

4-20: "Light attenuation was measured : : :" How specifically was light attenuation measured and with what instrument?

Response (4): The light intensity (I, µmol photons m-2 s-1) in underwater layers was measured in the field with Licor LI-192SB cosine-corrected light sensors connected to a Licor LI-185B quantum meter. Then the light extinction coefficient  $K_d$  and light distribution in the entire water column were calculated based on the Lambert-Beer Law,  $I_z = I_0 * exp(-z^*K_d)$ , where  $I_0$  and  $I_z$  are the light intensity at surface and depth z. We have added the information in the manuscript. Please see Page 4 Lines 25–27 in the "accept-changes" version of the revised manuscript.

4-25: "We used the measured values : : :" I don't quite understand this sentence, consider revising.

Response (5): "the measured values" have been changed to "the measured primary production data". Please see Page 4 Line 32 in the "accept-changes" version of the revised manuscript.

5-510: What weather forcing was used, specifically, and how was surface irradiance specified?

Response (6): We used atmospheric forcing including surface irradiance calculated from a downscaled weather model HARMONIE with a horizontal resolution of 2.5 km produced by the Royal Dutch Meteorological Institute (KNMI).

5-15: I see in the equations, detritus sinking is also calculated, but perhaps mention that here as well.

Response (7): Thanks for the suggestion. The sinking of detritus is added here. Please see Page 5 Lines 18–20 in the "accept-changes" version of the revised manuscript.

5-30: From what I can tell the bivalve biomass is constant, but perhaps clarify that here. Are the bivalves growing and dying or are they constant in time?

Response (8): Bivalves are growing and excreting nitrogen following the governing equation (22) in Table 1. However, our model does not account for the shellfish harvest mortality, occurring mostly in late summer. We have considered it as one of

the limitations of the model and discussed it in the first paragraph of Discussion. Please see Page 9 Lines 10–11 in the "accept-changes" version of the revised manuscript.

6-20:25: It would be useful to show some kind of climatology of the measurements with a window or errorbars that show the inter-annual variability. See figs in Testa, J. M., Murphy, R. R., & Brady, D. C. (2018). Nutrient-and climate-induced shifts in the phenology of linked biogeochemical cycles in a temperate estuary. Frontiers in Marine. https://www.frontiersin.org/articles/10.3389/fmars.2018.00114

Response (9): Thanks for the suggestion. We have replaced Fig. 5 with a climatology graph and updated the text accordingly.

9-20: Does the decreasing depth (presumably) also cause the benthic-pelagic coupling to become stronger? Are there bivalves in the more seaward stations but because there is a greater volume of water the grazing pressure just is less, on an areal basis?

Response (10): Good point. Yes, indeed. In a recent study about the spatial variability of tides in the Oosterschelde (Jiang et al., 2020), we found that the average depth and cross-sectional area decreases landwards (Fig. R1). This geometric feature induces tidal convergence, i.e., larger tidal amplitude at the landward end. Therefore, shallower water depth and stronger tidal mixing can contribute to stronger benthic pelagic coupling and higher benthic grazing pressure in the east of the basin. We have added this point here and in Discussion. Please see Page 9 Lines 17–20 and Page 10 Lines 16–17 in the "accept-changes" version of the revised manuscript.

**Reference**

Jiang, L., Gerkema, T., Idier, D., Slangen, A. B. A., and Soetaert, K. E.: Effects of sea-level rise on tides and sediment dynamics in a Dutch tidal bay, Ocean Sci., 16, 307–321, https://doi.org/10.5194/os-16-307-2020, 2020.

---

## Author Response (AR1)

Tom Fisher (Referee)
fisher@umces.edu

This is a well-written, multi-disciplinary manuscript addressing the distribution of phytoplankton and primary productivity in a Dutch estuary. The three approaches (observational, modeling, remote sensing) provide a strong basis for describing phytoplankton distributions and the causes of the patterns. The literature synthesis at the end of the manuscript sets the results of this manuscript in global context. Overall, a strong addition to the estuarine literature.

Response (1): We appreciate the referee's efforts in reading our manuscript and positive comments. The manuscript has been revised based on the following suggestions.

I suggest that the authors address a few issues that I think are missing: 1. Light limitation of phytoplankton growth is common in estuaries and often occurs in turbid, nutrient-rich, low-salinity waters with no vertical stratification. No data are presented on salinity or density in this manuscript, and the authors simply state that there is no stratification, citing another paper. They could be right, and water depth may limit vertical mixing. However, my experience suggests that vertical stratification in spring under high river flow conditions initiates the spring bloom. Even if there are no CTD data available to calculate vertical variations in density, the authors should at least mention the possibility that the low salinity areas with high nutrients and turbidity may be light-limited regions of the estuary.

Response (2): Thanks for the comment. The Oosterschelde used to be a coastal plain estuary, but not any more since the safety-oriented Delta Works in the late 1980s (https://en.wikipedia.org/wiki/Delta_Works). Many dams and sluices were built at approximately the same time cutting the freshwater input of the Oosterschelde, which became isolated from other delta networks (Ysebaert et al., 2016). The overall freshwater inflow into the bay is below 10 $m^3$ $s^{-1}$ (Ysebaert et al., 2016). For example, it was 3.2 $m^3$ $s^{-1}$ and 4.5 $m^3$ $s^{-1}$ in 2009 and 2010, respectively (Rijkswaterstaat data). This is a negligible amount compared to the flushing of the basin by tidal exchange, which is ~2 × $10^4$ $m^3$ $s^{-1}$, estimated from a typical tidal prism of 9 × $10^8$ $m^3$ in a 12-h tidal cycle.

Due to the greatly reduced freshwater input, the post-barrier Oosterschelde is well mixed most of the time. Based on our CTD casts in spring and summer, the surface-to-bottom salinity difference in a 10-m water column is below 0.1 psu (Figs.

R1 and R2).

Turbidity in the Oosterschelde does not resemble the typical distribution in an estuary. The suspended matter concentration is highest near the bay mouth and lowest near the northern branch where the aforementioned limited amount of freshwater enters (Wetsteyn and Kromkamp, 1994). In other words, the largest source of suspended sediment is the North Sea, rather than the landward end.

The Westerschelde, south to the Oosterschelde, is a true estuary with large freshwater and terrestrial of suspended sediment input, secchi depth of 0.2–2 m and salinity ranging 0 to 30. In contrast, the Oosterschelde is featured by greater transparency (secchi depth 3–5 m) and marine salinity conditions (salinity 30–33). Primary production in the Oosterschelde is much more limited by grazing and marine nutrient sources as discussed in the manuscript than light. Data in this paragraph are unpublished and measured by Jacco Kromkamp. We have clarified it in Section 2 (Page 3 Lines 23–25 in the "accept-changes" version) of the revised manuscript.

[Figure]

**Figure R1: Observed (a) water level and (b-f) salinity and temperature at OS2. The station location is showed in Figure 2. The observational periods during flood (6 March 2018) and ebb (8 March 2018) tides are marked with blue and red dotted lines, and the corresponding observational data are shown in the lower left and right panels, respectively.**

[Figure]

**Figure R2: Observed salinity and temperature at OS7 during one full tidal cycle on 4 June 2019. The station location is showed in Figure 2. The water level is shown with the dotted line.**

**References**

Wetsteyn, L. P. M. J., and Kromkamp, J. C.: Turbidity, nutrients and phytoplankton primary production in the Oosterschelde (The Netherlands) before, during and after a large-scale coastal engineering project (1980–1990), Hydrobiologia, 282/283, 61–78, http://doi.org/10.1007/BF00024622, 1994.

Ysebaert, T., van der Hoek, D. J., Wortelboer, R., Wijsman, J. W., Tangelder, M., and Nolte, A.: Management options for restoring estuarine dynamics and implications for ecosystems: A quantitative approach for the Southwest Delta in the Netherlands, Ocean Coast. Manage., 121, 33–48, https://doi.org/10.1016/j.ocecoaman.2015.11.005, 2016.

2. I am surprised that there are no data presented on salinity, temperature, river discharge, and river nutrient concentrations. The authors nicely show that advective inputs of shelf nutrients and phytoplankton is likely to be small, but never explore the role of river inputs. This could be a whole other paper, but they could at least mention that riverine inputs, both freshwater and nutrients, are likely driving the spring bloom. They could do additional model runs with half of the river discharge or half of the river N concentrations, but this might be more work than reasonable. I suspect that vertical stratification in the inner half of the estuary allows algal biomass to accumulate following high winter-spring river discharge. What would the model show if freshwater flow and/or river nutrient concentrations were halved? Can any of the temporal variations in the spring bloom be related to river flow?

Response (3): Thanks for the suggestion. The data on salinity, temperature, and river discharge is presented in the last response. The river nutrient is not significantly different from that in the Oosterschelde.

We conducted sensitivity test on the river discharge as suggested. A model run switching off the river discharge is compared with the baseline run. As we can see from the results (Figs. R3 and R4), dissolved inorganic nitrogen and chl-a are slightly higher in the baseline simulation including freshwater input, but the difference is minimal. The impact of freshwater is visible at OS5 but cannot reach the mainstem station OS2. These findings verify that freshwater input does not play a dominant part in the phytoplankton distribution in the Oosterschelde, as mentioned in the last response.

As readers may similarly wonder about the role of freshwater input, we have briefly added the above outlined explanation in Section 3.2 of the revised manuscript (Page 6 Lines 4–6 in the "accept-changes" version).

[Figure]

**Figure R3: Comparison between modeled and observed dissolved inorganic nitrogen (DIN) in 2009 at stations(a) OS5, (b) OS4, and (c) OS2. The panels are arranged based on their respective distance from the freshwater source. See Figure 2 for station locations. The two model scenarios include the baseline scenario and switching off freshwater input.**

[Figure]

**Figure R4: Comparison between modeled and observed chlorophyll-a in 2009 at stations (a) OS5, (b) OS4, and (c) OS2. The panels are arranged based on their respective distance from the freshwater source. See Figure 2 for station locations. The two model scenarios include the baseline scenario and switching off freshwater input.**

3. I made a few minor grammatical or wording suggestions to the pdf of the text and for improvements in the figures that will be easy to address. This isn't a long manuscript, and the above two issues can probably be addressed briefly in 1-2 pages.

Please also note the supplement to this comment:
https://www.biogeosciences-discuss.net/bg-2020-40/bg-2020-40-RC1-supplement.pdf

Response (4): The supplement is well received. Thanks for the minor suggestions, which are addressed as follows.

Page 4 Line 3-4: Not clearly stated. Bivalve grazing must also be burying and/or denitryfying phytoplankton N and P if grazing decreases prim prod. With no N and P losses, chla might decrease but primary production could be the same due to higher turnover.

Response (5): By grazing, bivalves remove N and P from the water column and keep the phytoplankton biomass low. On the other hand, bivalves release a smaller amount of inorganic nutrients into the water column by excretion and respiration, which may stimulate phytoplankton grown in the nutrient-limited summer months. We have removed the second half of the sentence for clarification. Please see Page 4 Line 9 in the "accept-changes" version of the revised manuscript.

Page 5 Line 10: validation? error rate or model accuracy? ok, I see it on p7. Add a sentence to Methods.

Response (6): We have changed it to "validation" and added this to the manuscript addressing the validation of the FABM model. Please see Page 5 Line 15 in the "accept-changes" version of the revised manuscript.

Page 5 Line 15: provide citation on sinking rate.

Response (7): We have added a reference (Eppley et al., 1967) here. Please see Page 5 Line 20 in the "accept-changes" version of the revised manuscript.

**Reference**
Eppley, R. W., Holmes, R. W., and Strickland, J. D. H.: Sinking rates of marine phytoplankton measured with a fluorometer, J. Exp. Mar. Biol. Ecol., 1, 191–208, https://doi.org/10.1016/0022-0981(67)90014-7, 1967.

Page 5 Line 26: specify weight ratio or give units. Also give a general range of PO4 concentrations to justify the N based model. Phytoplankton can be P and light-limited in the fresher parts of estuaries. Light limitation often occurs in the turbid 0-5 psu range, but no information seems to be available on vertical density and may not be available.

Response (8): Thanks for the comment. The unit of Chl:N ratio is mg Chl mmol $N^{-1}$, and the value (2) is prescribed based on the estimation of local species by Soetaert et al., 2001. Please see Page 5 Line 31 in the "accept-changes" version of the revised manuscript.

The $PO_4$ concentration ranges 0–2 mmol $m^{-3}$ (Fig. R5). Most time of the year, phosphorus is not limiting, except for a short period after the spring bloom. We have mentioned it in the Discussion that not including P limitation may result in the underestimation of DIN in this period. Please see Page 9 Lines 6–10 in the

"accept-changes" version of the revised manuscript.

As described in Responses (2) and (3), the freshwater inflow is extremely low. The turbid fresh (0-5 PSU) region and strong vertical salinity gradients hardly exist in the Oosterschelde.

[Figure]

**Figure R5: Time series of phosphate concentration during 1995–2013 at NIOZ stations OS1, OS3, and OS8.**

**Reference**

Soetaert, K., Herman, P. M., Middelburg, J. J., Heip, C., Smith, C. L., Tett, P., and Wild-Allen, K.: Numerical modelling of the shelf break ecosystem: reproducing benthic and pelagic measurements, Deep-Sea Res. Pt. II, 48, 3141–3177, http://doi.org/10.1016/S0967-0645(01)00035-2, 2001.

Page 6 Line 21: winter phytoplankton blooms can occur in estuaries at low temperatures. Vertical stratification, sometimes defined by <0.5 psu, is the dominant control by limiting the depth of mixing in turbid waters especially with high FW flows.

Response (9): This comment also relates to our unclear description of the limited freshwater contribution. The Oosterschelde is not a typical estuary with high freshwater input. We have made the clarification here. Please see Page 6 Lines 26–27 in the "accept-changes" version of the revised manuscript.

Page 6 Lines 29-30: Was there any relationship between peak or integrated biomass and total river flow into the estuary? I'm guessing that the big or sustained peaks are positively associated with river discharge during the bloom period.

Response (10): Yes, the phytoplankton biomass at station RWS1 (at the mouth of the Westerschelde Estuary) is mostly influenced by the discharge of the Westerschelde. But for the most parts of the Oosterschelde, as presented in the manuscript, grazing pressure is the dominant control on phytoplankton biomass.

Page 8 Lines 22-25: Mention in methods that this issue is addressed in Discussion.

Response (11): OK. This relates to Response (8).

Page 8 Line 31: add a little more detail on how this was calculated here.

Response (12): Thanks for the suggestion. There are many ways to calculate residence time, and it is necessary to indicate the calculation here. The residence time is estimated by two methods in a model tracer experiment (Jiang et al., 2019). Briefly, each grid cell is filled with tracer and these two methods quantify the decay rate of tracer. The first method integrates the remnant function to calculate residence time

$T_r = \int_0^\infty C(t)/C_0 dt$, where $C(t)$ and $C_0$ are the instantaneous and initial tracer

concentration in each grid cell. The second method quantifies the time when $C(t) = e^{-1} C_0$, since the tracer concentration decreases exponentially in a well-mixed system. Based on our estimate, these two methods result in similar residence time in our system. We have added these two methods to the sentence. Please see Page 9 Lines 17–18 in the "accept-changes" version of the revised manuscript.

**Reference**

Jiang, L., Soetaert, K., and Gerkema, T.: Decomposing the intra-annual variability of flushing characteristics in a tidal bay along the North Sea, J. Sea Res., 101821, https://doi.org/10.1016/j.seares.2019.101821, 2019.

Page 9 Line 21: but nutrients are probably increasing towards the river end member due to light limitation of algal growth in the inner estuary.

Response (13): Because of the "missing" or weak river end member, the spatial gradients of nutrients and turbidity are not as strong as those in typical estuaries. Thereby, we differentiate the Oosterschelde, representing coastal bays with limited freshwater input but dominant marine influences, from river-dominated systems and other types in Section 6.

Page 9 Line 32: substantial land-derived nutrients, and light limitation of phytoplankton in turbid, low salinity areas of the estuary.

Response (14): This comment also relates to our unclear description of the limited freshwater contribution in the Oosterschelde. Because of that, the largest nutrient source is the marine import, contributing to the seaward increasing chl-a distribution.

Page 10 Line 10: is maintained

Response (15): Corrected. Please see Page 11 Line 9 in the "accept-changes" version of the revised manuscript.

Page 11 Line 26: trophic levels.

Response (16): Thanks for the correction. This sentence is rephrased. Please see Page 12 Line 27 in the "accept-changes" version of the revised manuscript.

Page 11 Line 28: This is barely addressed. Either add likely climate effects to Discussion or remove this statement here and in Abstract.

Response (17): Thanks for the suggestion. The climate effect is deleted here and in Abstract. Please see Page 12 Lines 22–28 and Page 1 Lines 24–25 in the "accept-changes" version of the revised manuscript.

Figure 5: The time axis of Figs 4 and 5 is not sufficiently clear to tell when the blooms occur. They said it was spring, but it would be easy to add month or half-year tics to show this more clearly.

Response (18): Thanks for the suggestion. We have changed the interval of grid lines to two months in Fig. 4, and Fig. 5 has been replaced with a monthly average graph.

Referee 2
This is one of the most well-written papers I have reviewed and everything was well laid out and easy to follow. However, I was left with wanting a lot more information and discussion from the authors. This is the first time I have ever said this, but this paper is too short. There are three different approaches used in the paper, which is a plus, but there is a lot of information that could be shared about each approach. The main conclusions for the paper are clear and well supported, but this is a case study, and I think there should be a lot more acknowledgement and discussion of the nuances in the variability of the spring bloom in the Oosterschelde; it has not always followed the described pattern. More specifics are provided below:

Response: The referee's constructive suggestions are much appreciated. We have revised the manuscript and provided the suggested information. Please see the following point-by-point replies.

General Comments
1. More information and discussion of the field data
(a) Page 6, line 17-20: There is some discussion in here about how nutrients, light, and temperature effect the phytoplankton biomass annual cycle, but none of the data presented demonstrate or support these claims. What is this based off of, other people's findings or the authors own analysis? If these are the authors own conclusions, then I would like to see data and analysis to support these claims.

Response (G1a): Thanks for the comment. We have added the model results of temperature, nutrients, and light factors affecting the growth rate (Fig. 8) and explained the seasonal controls of phytoplankton variability in Section 4.2 (Page 7 Lines 19–29 in the "accept-changes" version of the revised manuscript). In winter, the low chl-a is a result of low temperature and light conditions that improve in March and April and initiates the spring bloom. The bloom algae consume nutrients and lead to post-bloom nutrient limitation. In late summer and early fall, light, nutrients, temperature still fuel high growth rate, while the low biomass results from grazing. When temperature and light start constraining primary production, the chl-a decreases and nutrients accumulates, entering the next annual cycle.

(b) A figure of DIN concentrations, similar to figure 4, would be beneficial.

Response (G1b): Thanks for the suggestion. We have plotted the 19-yr timeseries of

DIN. DIN at the offshore station RWS2 is consistently lower than RWS1 and other inner-bay stations, while the stations inside the Oosterschelde show little difference in DIN concentration from each other (Fig. R1). Thus, the DIN spatial heterogeneity in the bay is not as strong as chl-a (Fig. 4). Given that the DIN annual cycle is presented in Fig. 6, Fig. R1 is not included in the manuscript.

[Figure]

**Figure R1: Time series of DIN (dissolved inorganic nitrogen) concentration during 1995–2013 at (a) NIOZ stations OS1, OS3, and OS8 and (b) Rijkswaterstaat stations RWS1–RWS4. Intervals between grid lines indicate two months. See Figure 2 for station locations.**

(c) Why is OS6 not included in figure 2?

Response (G1c): OS6 is to the north of OS5 and not in the model domain. Thus, OS6 is not presented in the map.

(d) Page 4, line 16-17: More information on the 14CO2 uptake experiments would be helpful - Who did the experiments? At all stations? For all sample dates in 2010? Is this data published elsewhere?

Response (G1d): One of our co-authors Jacco Kromkamp did the experiments. The measurements were done at stations OS2 and OS8, as shown in Fig. 7, and yes, for all sample dates in 2010. The data is not published before. We have added the suggested information to Section 3.1 and Author contributions. Please see Page 4 Lines 21–22 and Page 13 Lines 8–9 in the "accept-changes" version of the revised manuscript.

(e) The SD bars in figure 2 are large for both gradients, suggesting a wide range of [chla] at all stations during spring between 1995 and 2013. The average values show a tendency towards higher [chla] at the mouth, but the large SD demonstrate a lot of inter-annual variability. Based on figure 4a, it appears that the pattern in spring phytoplankton biomass described in this paper dominated between 2000 and 2009. Pre-2000, [chla] at the head and mid-bay repeatedly matched or surpassed the mouth, 1998 being the clear exception. After 2009, [chla] during spring appears to become less distinct between each location. This does not negate the conclusions of the paper, but the years that do not match the pattern should be acknowledged. It is not expected that every year will always be the same, so what might have happened in the years that didn't follow the pattern?

Response (G1e): Good point. We agree with the referee that the spatial chl-a gradient shown in Fig. 2 is not universal in every year in Fig. 4. The gradient is obvious overall (Fig. 5), especially in some years (such as in 2005 and 2008 in Fig. 4a) and not so much in other years (such as in 1997 and 2012 in Fig. 4a). There are several potential causes.

Firstly, due to shallowness and being surrounded by land, water in the bay head is heated up slightly faster than that in the North Sea, so the spring bloom sometimes occurs several days earlier. If the sampling activity happens during the bloom in the North Sea, the chl-a distribution shows a seaward increasing trend. However, if the measurement is undertaken during the bloom in the bay, the bay head may exhibit higher phytoplankton biomass. For example, the peak biomass at the seaward end occurred later than at the landward end in Years 1996, 1997, and 2005 (Fig. 4b), whereas the spring bloom of the entire basin happens almost at the same time in Years 2002 and 2008 (Figs. 4a and 4b). Moreover, the chl-a concentration tends to be slightly higher in the landward end in February (Fig. 5). Thus, the interannual variability of the spring bloom timing can contribute to changes in the spatial chl-a gradient.

Secondly, the sampling frequency is usually biweekly or even monthly, so one data point in the time series may miss the peak bloom or result in different spatial chl-a

distribution. For instance, in 2010, the overall magnitude of the spring bloom is relatively low, and the spatial chl-a gradient is more obvious in the RWS data than in the NIOZ data due to different sampling dates (Figs. 5a and 5b).

Given the above concerns, the seaward increasing chl-a gradient may not be discernible in one year's data. Therefore, we present the gradient with the average biomass in March to May during 19 years (Fig. 2), as well as by month (Fig. 5). The large standard deviations in Figs. 2 and 5 do include the interannual variability as the referee suggests. Note that the phytoplankton biomass before, during, and after the spring bloom also creates large variability in three months, which is another source of the presented standard deviations. We have added a paragraph in Discussion including these effects of temporal variability on the spatial phytoplankton distribution. Please see Page 10 Lines 3–13 in the "accept-changes" version of the revised manuscript.

2. More information and discussion of the model results (a) Figure 6, 7, 9: Are the field observations in these figures averages? If there is any way to calculate the standard deviation for these values, it should be included.

Response (G2a): The mesotidal Oosterschelde is well-mixed vertically (see the CTD casts of temperature and salinity at OS7 in Fig. R2 for example). For DIN and chl-a observations shown in Figs. 6 and 9 (now Fig. 10), the NIOZ and Rijkswaterstaat monitoring programs use the near-surface sample to represent the entire water column. Thus, unfortunately, there is no three measurements or replicates at each station to calculate the standard deviations.

For the *NPP* data in Fig. 8 (now Fig. 9), light attenuation was measured once during the cruise and the overall primary production is an integration of the entire water column. That is, we have only one *NPP* result for each station on each sampling date and are unable to calculate the standard deviations for *NPP*.

[Figure]

**Figure R2: Observed salinity and temperature at OS7 during one full tidal cycle on 4 June 2019. The station location is showed in Figure 2. The water level is shown with the dotted line.**

(b) Page 7, line 12: The authors mention that before the bloom, phytoplankton biomass and growth rates were low, but this data is not really presented. I know [chla] = phytoplankton biomass, so maybe keep the terminology consistent in the paper. However, the growth rate data from the model is something that I think should be presented, it sounds like it is interesting.

Response (G2b): Thanks for the comment. Primary production is the product of phytoplankton biomass and growth rate. In our model formulation, growth rate is a function of temperature, nutrient, and light factors, as shown in Equation (2) in Table 1. We have extracted these three factors from the model (Fig. 8). Their product should

be proportional to the growth rate and be used to denote the relative changes in the growth rate (Fig. 8). The discussion based on biomass and growth rate has been updated with the new figure included. Please see Page 7 Lines 19–29 in the "accept-changes" version of the revised manuscript.

(c) Page 7, line 17-18: I wanted more detail here, rather than saying NPP is generally higher at OS8 compared to OS2. The authors note that the model overestimates NPP at OS8 in the fall 2010, but that overestimation makes it difficult to compare the two sites in figure 7. What is the average + SD observed NPP at each station? Is it significantly different? On average, how much higher is NPP at OS8 compared to OS2 in observed and modeled data?

Response (G2c): Thanks for the suggestion. The observed *NPP* at OS8 and OS2 is $902.6 \pm 928.4$ mmol C m$^{-2}$ d$^{-1}$ and $722.5 \pm 794.6$ mmol C m$^{-2}$ d$^{-1}$, respectively. Although the observed NPP at OS8 is slightly higher, their difference is not significant according to the t-test ($t = 0.59$, $p > 0.05$, $n = 16$). In contrast, the modeled *NPP* at OS8 ($1033.9 \pm 1084.3$ mmol C m$^{-2}$ d$^{-1}$) is significantly higher that OS2 ($606.0 \pm 499.5$ mmol C m$^{-2}$ d$^{-1}$) according to the t-test ($t = 6.85$, $p < 0.05$, $n = 365$). We have added these statistical analyses to the revised manuscript. Please see from Page 7 Line 29 to Page 10 Line 2 in the "accept-changes" version of the revised manuscript.

(d) Figure 7: The authors explain the overestimation at OS8, but why did the model miss the highest peak in NPP at both stations around day 175?

Response (G2d): Good question. The modeled *NPP* is influenced by temperature, nutrients, light, and phytoplankton biomass. Light and temperature is reasonably simulated by the hydrodynamic model (Jiang et al., 2019), and the slight deviation of simulated chl-a (Day 540 in Fig. 6) cannot fully explain the underestimation of *NPP*, which, thus, is related to nutrients. Fig. 7 show that the underestimation of *NPP* actually starts right after the spring bloom (Day 125) and lasts till around Day 175. This is a period that DIN is underestimated (Days 490–540 in Fig. 6). As discussed in the manuscript, this results from our N-based model excluding the effect of phosphorus or silicon limitation. Therefore, DIN is depleted too early and thus limiting *NPP* (Fig. 8), which should decline more slowly or even increase slightly (Fig. 7). We have added these discussions in Sections 4.2 and 5. Please see Page 7 Line 27 and Page 9 Lines 6–10 in the "accept-changes" version of the revised manuscript.

Response (S3): Thanks for the comment. We agree that the manuscript focuses on the spatial pattern, which cannot be discussed without temporal variations. These two sentences have been rephrased to emphasize the temporal changes and natural and anthropogenic controls on phytoplankton distribution. Please see Page 12 Lines 22–28 in the "accept-changes" version of the revised manuscript.

Referee 3
This paper describes a coupled observational, modeling and satellite observational study of an estuarine system in the North Sea. Overall, the story and results were well conveyed and the conclusions regarding drivers of spatial and temporal variability in the estuary were supported. The main take away is that there is a Type I phytoplankton distribution and it is mainly driven by benthic grazing pressure in the landward stations. The model supports the importance of grazing pressure on the spatial distribution by numerically removing bivalves in the modeling system. Modeling estuarine primary production and chl-a distribution can be particularly challenging, and I think the author's did a pretty good job at capturing overall NPP magnitude and some of the temporal variability, compared to 14-C NPP incubation data. The synthesis at the end is particularly useful, especially related to the discussion of how different mechanisms can lead to similar patterns of chl-a distribution, depending on the system.

Response (1): Thanks for the positive feedback and the following suggestions. We have revised the manuscript as suggested and replied to the comments point by point.

The main methodology and results that need to be improved upon, or omitted, relates to the use of the satellite observations. The author's use one image (Fig. 10) and it doesn't really track with the results and conclusions of the rest of the paper. In fact, the chl-a concentration is highest in the landward stations where in most observations showed lower chl-a concentration. I understand the desire to do this coupled methodological approach, but in my opinion if satellite data is to be used, it should be developed a bit more to support the observational and modeling work. There is definitely a lot of value in using these data, but acquiring more spring bloom images from MERIS data that fall within the observational window would offer a bit more support for the other results.

Response (2): Thank you for the suggestions.

The Envisat MERIS satellite data can be used to retrieve chl-a concentrations at a spatial concentration of ca 300 m (FR, full resolution data) to ca 1 km (RR, reduced resolution data). Their spatial resolutions are typically not sufficiently high for the application of the Oosterschelde, because the narrowest portions of the basin and the northern branch are around 3 km , and the Oosterschelde has intertidal flats that fall

dry during low water (Fig. R1). Therefore, the land or intertidal pixels may interfere with the MERIS data in the Oosterschelde region.

van der Woerd et al. (2011) have investigated the surface chl-a in the North Sea with MERIS reduced resolution output. Their processed MERIS satellite data show that results within the Oosterschelde should be treated with caution, especially at narrow regions (Fig. R2). In contrast, the 10-m-resolution Sentinel-2 MSI data applied in our study have a spatial resolution suitable for the Oosterschelde.

In addition to the demand of a high spatial resolution, the satellite data to be used in our study also needs to be taken at high tides. One third area of the Oosterschelde is covered by intertidal flats and the water around the flats is extremely shallow (Nienhuis and Smaal, 1994). Bottom reflectance may become another source of errors at low tides (cf Arabi et al., 2019, for optically shallow water effects from MERIS full resolution images of the Wadden Sea). Availability of high tide, low cloud cover images that can show the overall spatial chl-a gradient in the Oosterschelde. are hence very limited.

Moreover, the satellite image in Fig. 11a offers, as a snapshot, valuable insight into the spatial gradient. The spatial phytoplankton pattern shown in Fig. 11a was also detected in the model output (Fig. 11b), which partly validates the model. Although the seaward increasing chl-a gradient is most common in the Oosterschelde in spring, it changes with time. That is, when discussing the spatial phytoplankton variability, we have to be aware of the temporal variability. When describing the general spatial gradient, the less general "exceptions" needs to be noticed. In the revised manuscript, we have emphasized the importance of temporal variability. The satellite image and the less frequent spatial gradient it displays (Fig. 11) fits in that standpoint. Thereby, we tend to retain Section 4.3 and Fig. 11a.

[Figure]

**Figure R1: The cross-sectional area and width of the Oosterschelde from the mouth to its eastern end. The northern branch (Fig. 1) is excluded from the calculation because of a different orientation of channels. This figure is Fig. 6 in Jiang et al., 2020.**

**Reference**

[revised manuscript text omitted]

---

## Author Response (AR2)

Associate Editor Decision: Publish subject to minor revisions (review by editor) (18 Jun 2020) by Carol Robinson

Comments to the Author:

Dear Authors

Many thanks for your revised manuscript and responses to the reviewers. This has now been re-reviewed. The second set of reviews are all positive but all have some suggestions for further improvement and clarification. I look forward to receiving your responses to these latest comments and your revised manuscript.

Best wishes
Carol Robinson

Response:

Dear Editor,

Thanks for handling our manuscript in the re-review process. All the comments from three referees are received and addressed. We have responded point by point as follows and are submitting the revised manuscript with this letter.

Best wishes,
Long Jiang
On behalf of co-authors

Report #1
Referee #2: Nicole Millette, nmillette@vims.edu

Overall, the authors put thought into my previous comments and either incorporated it into their paper or gave a detailed response. I do have some minor comments still I would like them to consider.

Response (1): Thanks for the positive comments on our revised manuscript. The minor comments are addressed point by point as follows.

Comments
• Page 4, line 13-15: At least provide a reference for where people can get more detailed information about the NIOZ sample collection and analysis.

Response (1): We have provided two references (Cloern and Jassby, 2010; Smaal et al., 2013) in which the NIOZ datasets were applied and one reference (Kromkamp and van Engeland, 2010) describing the phytoplankton sampling activities in detail. Please see Page 4 Lines 16–17 in the "accept-change" version of the revised manuscript.

**References**
Cloern, J. E., and Jassby, A. D.: Patterns and scales of phytoplankton variability in estuarine–coastal ecosystems, Estuaries Coast., 33:230–241, http://doi.org/10.1007/s12237-009-9195-3, 2010.
Kromkamp, J. C., and van Engeland, T.: Changes in phytoplankton biomass in the Western Scheldt Estuary during the period 1978–2006, Estuaries Coast., 33, 270–285, http://doi.org/10.1007/s12237-009-9215-3, 2010.
Smaal, A. C., Schellekens, T., van Stralen, M. R., and Kromkamp, J. C.: Decrease of the carrying capacity of the Oosterschelde estuary (SW Delta, NL) for bivalve filter feeders due to overgrazing? Aquaculture, 404, 28–34, http://doi.org/10.1016/j.aquaculture.2013.04.008, 2013.

• Page 4, line 21-22: As with the NIOZ sampling data, at least provide a reference for the method that was used to estimate primary production with 14C. However, since these experiments were done for this project, or at least this is the first time this data is being published, I would suggest that more information is needed than one sentence. What containers were used, what volume of water, how much water was used in analysis, how was analysis done, etc. This was information I was left wondering about.

Response (3): Thanks for the suggestion. The data measured in 2010 was not published elsewhere, but the $^{14}$C method has been routinely used in the measurement of primary production by NIOZ researchers. Details are described in two earlier papers (Kromkamp and Peene, 1995; Kromkamp et al., 1995) and presented in the

next paragraph. We have included these two references here pointing to details of the experiments. Please see Page 4 Lines 16–17 of the "accept-change" version of the revised manuscript. Please see Page 4 Lines 23–24 in the "accept-change" version of the revised manuscript.

The 50 ml surface water samples, to which 200 µL of 185 kBq mL$^{-1}$ $^{14}$C-NaHCO$_3$ (Amersham) was added, were incubated in a rotating incubator on board the RV Luctor immediately after sampling at *in situ* temperatures. The samples in the incubator were exposed to 10 different irradiances (excluding the dark bottles) up to a maximal irradiance of 810 µmol m$^{-2}$ s$^{-1}$. After incubation the samples were gently filtered over 0.45 µm nitrocellulose filters. The incubation period was 2 h. Filters were placed in HC1 fumes for 30 min, air dried and counted in a Beckmann LSC (LS5000TD). A scintillation cocktail was prepared using 0.5% (w/v) PPO (2,5-diphenyloxazol, p.a. Merck) in toluene (Baker, technical grade). Correction for quench took place using the shift in the Compton peak (H-number) according to the manufacturer's instructions. Dissolved inorganic carbon was determined by potentiometric titration. Primary production was calculated using an isotope discrimination factor of 1.05. The rate of carbon fixation in the dark bottles was subtracted from that in the light bottles in order to avoid overestimates of primary production by phytoplankton due to chemosynthetic processes.

**References**

Kromkamp, J., and Peene, J.: Possibility of net phytoplankton primary production in the turbid Schelde Estuary (SW Netherlands), Mar. Ecol. Prog. Ser., 121, 249–259, http://doi.org/10.3354/meps121249, 1995.

Kromkamp, J., Peene, J., van Rijswijk, P., Sandee, A., and Goosen, N.: Nutrients, light and primary production by phytoplankton and microphytobenthos in the eutrophic, turbid Westerschelde estuary (The Netherlands), Hydrobiologia, 311, 9–19, http://doi.org/10.1007/BF00008567, 1995.

• Figure 13: The authors have made it clear that they still want to include this figure in the paper, and none of the information that they present is wrong so I cannot say that it should not published. However, I would like to explain my concerns. I primarily have an issue with the figure and the naming of different types of [chla] spatial gradients, not the text of the synthesis section. I do think this figure is really interesting and useful, but to include it at the end of a paper about a single estuary is a disservice to the figure. What the authors have is the basis for a potentially interesting review paper on [chla] spatial patterns in estuaries-coastal systems and causes for these patterns. I would argue that it is not appropriate to introduce a new scientific categorization at the end of an original research paper. It is fine to compare spatial patterns in Oosterschelde and what is driving these patterns to other systems, as the authors do in the text, as a way to demonstrate the wider context on these findings. However, the authors decide to go a step further and classify the different types of spatial gradients (as type I-V). My opinion is that if you are presenting a new

categorization for scientists to use, then more effort is needed to explain and demonstrate each different "type". Most of the "types" described here are in relation to Oosterschelde, and not as their own type. Again though, I acknowledge that this is just my opinion and I cannot challenge any of the information they present in the figure. I just think this figure and the categorization is a step too far for this paper.

Response (4): Thanks for expressing the concerns about Fig. 13 and its interpretation in the text. We agree with the referee that the Synthesis section and Fig. 13 should be closely related to the original study of the Oosterschelde rather than develop into an independent review paper. In the recent revision, we have edited the sections Abstract, Synthesis, and Summary and replotted Fig. 13 to put them in closer context to the Ooscterschelde case. Specifically, we have weakened the idea that our study aims for a review and categorization (Types I-V) of global estuarine-coastal system, and strengthened our general implication that one or several dominant environmental drivers can shift the chl-a spatial distribution in an estuary or coastal bay, as found in our study.

Report #2
Referee #4: Koenraad Muylaert, koenraad.muylaert@kuleuven.be

I agree with the other reviewers that the paper is well-written and the results are convincing. I have no major comments. I only have a few remarks that the authors might take into consideration.

Response (1): Thanks for reviewing our manuscript and providing insightful and constructive comments. The following comments have been addressed correspondingly.

The main conclusion of this study is that the gradient in decreasing chlorophyll a concentration along the longitudinal axis of the embayment is caused by input of phytoplankton from the sea and grazing pressure exerted by bivalves in the inner bay. A similar conclusion was previously reached in another paper by Jiang et al that explored the fate of seston in the Oosterschelde bay. The difference between this and the previous paper is that this paper includes a primary production function (backed up by in situ primary production measurements). This begs the question: 'what is the relative importance of in situ primary production in the bay versus import from productive coastal waters'?

Response (2): A good question. Thanks for following our last paper. Based on the measured net primary production ($NPP$) and phytoplankton biomass (chl-a), the calculated phytoplankton turnover time in the Oosterschelde during warm months (March to October) is 0.92–5.2 days at OS2 and 0.13–4.3 days at OS8, which is much shorter than the residence time at most locations of the bay (Fig. R1). In other words, near the bay mouth, the import from the North Sea is on a similar order of magnitude to *in situ* production, while further into the bay, *in situ* primary production is more important than import. We have added the calculation in Methods, Results, and Discussion. Please see Page 5 Lines 7–8, Page 8 Lines 5–6, and Page 10 Lines 32–34 in the "accept-change" version of the revised manuscript.

[Figure]

**Figure R1:** Average residence time (days) in the Oosterschelde computed with a numerical model.

Fig 10 seems to suggest that import of phytoplankton from coastal waters is of little importance for chlorophyll a concentrations in the Oosterschelde bay, except for the station close to the mouth. Fig 12, however, gives a different impression and suggests that chlorophyll a concentration is almost halved along most of the bay when import is turned off. Or is import from coastal waters only important during the spring bloom? Some quantification of the relative importance of import versus local production in the bay during the course of the year would therefore be valuable.

Response (3): Thanks for the comment. The imported biomass can be important on two conditions: (a) a large difference in phytoplankton biomass in and out of the bay and (b) residence time sufficiently short compared to the phytoplankton turnover time as discussed in Response (2). From that perspective, Figs. 10 and 12 can quantitatively indicate the roles of local production versus import.

In Fig. 10, when halving the nutrient and phytoplankton loading from the coastal seas, the difference from the baseline case makes little difference, except in spring. That is, import from the North Sea is much more important in spring than in other seasons. Fig. 12b is plotted when the chl-a difference between the North Sea and Oosterschelde is largest of the year. The imported biomass largely depends on the magnitude and timing of the bloom in the North Sea. Fig. 12b also displays that the imported biomass decreased with distance from the bay mouth.

This is a follow-up comment of the last one, which helps us dig further into the seasonal and spatial variability of the importance of tidal import. We have improved our manuscript by adding the above discussion. Please see Page 10 Line 31 to Page 11 Line 4 in the "accept-change" version of the revised manuscript.

Presenting only chlorophyll data along a longitudinal gradient of an estuarine system can be misleading because the volume of the system generally decreases

exponentially from the seaward to the landward end. Therefore, what is happening near the mouth of the bay is quantitatively much more important than what is happening near the head of the bay.

Response (4): Thanks. Good point. We have considered the different cross-sectional area from the western to eastern ends, and added Fig. 12c to display the chl-a (biomass) integrated by cross-sectional area and volume. Results indicate the western Oosterschelde, as expected, accounts for a small surface area but a large volume. 60% of the phytoplankton biomass during the peak spring bloom is distributed in the area 0-10 km east to the bay mouth. Therefore, as the referee suggests, the effect of tidal impact is mostly limited to the western basin, but contributes largely to the overall biomass. We have added a paragraph to Discussion elaborating the quantitative analyses of the contributions of import from the North Sea. Please see Page 11 Lines 5–9 in the "accept-change" version of the revised manuscript.

It is hard to get an idea of the amount of water that is exchanged between the bay and coastal waters. The tidal prism is given, but this value is of little use if the total volume of the bay is not known. Would it be possible to add this information?

Response (5): Absolutely. The basin volume is 2.76 km$^3$ according to an early measurement (Nienhuis and Smaal, 1994), and our model calculation gives a similar number. This is about three times the tidal prism. The information is added to Section 2. Please see Page 3 Lines 25–26 in the "accept-change" version of the revised manuscript.

**Table R1: The surveyed bivalve biomass (kilotons fresh weight) in the Oosterschelde in the year 2009 (Data source: Wageningen Marine Research). This is adopted from Table 1 of Jiang et al. (2019).**

| Species | Scientific name | Eastern | Central | Western | Northern |
|---|---|---|---|---|---|
| Cockles | *Cerastoderma edule* | 4.62 | 13.64 | 13.97 | 8.41 |
| Blue mussels (wild) | *Mytilus edulis* | 0.22 | 0.00 | 0.00 | 0.23 |
| Blue mussels (cultured) | *Mytilus edulis* | 0.00 | 11.23 | 18.89 | 2.98 |
| Pacific oysters (wild) | *Crassostrea gigas* | 13.89 | 9.97 | 8.66 | 10.67 |
| Pacific oysters (cultured) | *Crassostrea gigas* | 6.71 | 0.00 | 0.00 | 0.00 |
| Baltic clams | *Limecola balthica* | 0.03 | 0.07 | 0.16 | 0.02 |
| Manila clams | *Venerupis philippinarum* | 0.30 | 0.00 | 0.00 | 0.00 |
| Razor clams | *Ensis leei* | 0.22 | 0.05 | 8.14 | 0.91 |
| Soft-shell clams | *Mya arenaria* | 0.01 | 0.01 | 0.01 | 0.01 |

The model is only very summarily described in the methods section. It would be interesting to add some general information on how N, light and temperature effects

on primary production were modeled, and how grazing and nutrient cycling by bivalves were modeled.

Response (8): Thanks for the suggestion. The referee mentions several key processes in our model.

The nitrogen and light effects on primary production follow the Monod equation, and the temperature effect is represented by $Q_{10}$ coefficient (Equation 2, Table 1). The grazing by bivalves is modeled as the product of species-specific grazing rates, temperature effects, bivalve biomass, and the concentration of overall particular organic matters including phytoplankton, zooplankton, and detritus (Equation 11, Table 1). The recycled nutrients are quantified by bivalve excretion, respiration, and remineralization of feces and pseudofeces that enters the Benthic Detritus pool (Equations 12, 13, 16, and 20, Table 1 and two arrows in Fig. 3).

We plotted each modeled process in Fig. 3, which is calculated using the equations in Table 1 and parameters in Table 2. These processes are self-explanatory. Maybe it was a bit unclear. In the revised manuscript, we have directed the readers caring about all processes in the NPZD model to these tables and figures. Please see Page 5 Lines 29–30 in the "accept-change" version of the revised manuscript.

I agree with the previous reviewers that the remote sensing data add very little to the conclusion of the study. I think the statement in the abstract 'satellite data substantiate the roles of benthic grazing and tidal import' is not supported by the spatial distribution of chlorophyll a concentration shown in the satellite image.

Response (9): Thanks for the comment. We have changed this sentence to "Satellite data captured a post-bloom snapshot indicating the temporally variable phytoplankton distribution". Please see Page 1 Lines 21–22 in the "accept-change" version of the revised manuscript.

How important is nutrient remineralisation by bivalves for supporting primary production in the bay? That nutrient recycling by bivalves is important is briefly mentioned on p. 9 l. 27, but it is not quantified, nor supported by data.

Response (10): Thanks for the questions.

Maybe it was not clear in the text, but we quantified and analyzed the effect of the remineralized nutrients on phytoplankton growth in the second paragraph of Section 4.2. The regenerated nutrients, calculated with the methods elaborated in Response (8), fuel the high growth rate and primary production after the spring bloom. Specifically, in Fig. 8, the growth rate was decomposed into three factors: temperature, light, and nutrients. Around Day 125, the spring bloom is terminated due to nutrient limitation (red line, Fig. 8). After that, even though DIN and chl-a remain at a relatively low

level (Fig. 6), the growth rate is even higher than during the spring bloom (orange dashed line, Fig. 8), which is related to the high temperature (black line, Fig. 8), as well as nutrients (red line, Fig. 8). DIN concentrations are also low in the North Sea in summer, so the DIN maintaining high production here is a result of remineralization. However, despite a high grow rate, the phytoplankton biomass is still low (Fig. 6). This means that the stimulated production by nutrient recycling is not as much as that grazed by bivalves, which leads to our discussion referee mentions (used to be on Page 9 Line 27, now on Page 10 Lines 1–2 in the "accept-change" version of the revised manuscript.). We have cited figures support the claim.

Additionally, we also mentioned that the overestimate of recycled nutrients at OS8 by our model (Days 600–650 in Fig. 6a) resulted in the overestimate of *NPP* (Days 235–285 in Figs. 7a and 8a) in Section 4.2.

The "synthesis" that gives a kind of classification of estuaries and bays is very interesting but it too short to be of much value. This could be very interesting if it were elaborated in a review paper that includes an in-depth discussion on the relative importance of different processes that control phytoplankton in estuaries. The "synthesis" also includes much information that is not relevant for this study. It is a classification of estuaries while the Oosterschelde is not really an estuary but a coastal bay. Because so much of the discussion deals with other systems, the comparison with similar systems where phytoplankton distribution is governed by bivalve filtration is too brief. This phenomenon has been reported in other estuaries and many of the other systems where bivalves control spatial distribution of phytoplankton are not mentioned (e.g. Willapa Bay and San Francisco Bay). It would be valuable to have more in-depth discussion of the conditions that lead to such a situation of strong top-down control by bivalves.

Response (11): Thanks for the comments that include multiple points.

Firstly, the synthesis section read like a separate review paper. In the recent revision, we have edited the sections Abstract, Synthesis, and Summary and replotted Fig. 13 to put them in closer context to the Ooscterschelde case. Specifically, we have weakened the idea that our study aims for a review and categorization (Types I-V) of global estuarine-coastal system, and strengthened our general implication that one or several dominant environmental drivers can shift the chl-a spatial distribution in an estuary or coastal bay, as found in our study.

Secondly, it is a classification of estuaries, while the Oosterschelde is not an estuary. Maybe there was confusion in our manuscript, we actually looked at both estuaries and coastal bays and referred to them as estuarine-coastal systems. The estuarine-coastal systems include estuaries, bays, lagoons, fjords, river deltas, and plumes and have many characteristics in common, as described in the second sentence of Introduction. We also included several coastal bays in Synthesis, such as the

Willapa Bay and Tomales Bay.

Thirdly, we have added two paragraphs in Discussion on bivalve top-down control on phytoplankton. Please see Page 10 Lines 5–25 in the "accept-change" version of the revised manuscript. The new discussion includes the composition of bivalves (natural, commercial, and invasive species, as mentioned in Response (6)), the impact on the pelagic foodweb, assessment of carrying capacity, and a comparison of filtration capacity combined with previous papers (Jiang et al., 2019; Smaal and van Duren, 2019). The suggested case studies in Willapa Bay and San Francisco Bay have been included in the discussion.

Response (6): SPM results were excluded from our study. We have removed this sentence and focused on the description of Chl-a here. Now on Page 6 Lines 15–16 in the "accept-change" version of the revised manuscript.

p7, line 13: CC is not commonly used. I think the authors mean r2?

Response (7): The correlation coefficient ($CC$) is the square root of $r^2$. We have changed it to $r^2$. Now on Page 7 Line 16 in the "accept-change" version of the revised manuscript.

p7, lines 18 to end of paragraph: insert "predicted" or "modeled" every few sentences to remind readers that these are model results, not observations.

Response (8): Thanks for the suggestion. This is addressed. Now this paragraph is located from Page 7 Line 22 to Page 8 Line 6 in the "accept-change" version of the revised manuscript.

p8, line 16: "Oosterschelde MODEL (Fig. 7)."

Response (9): "model" added. Now on Page 8 Line 20 in the "accept-change" version of the revised manuscript.

p8, line 18: "Therefore, tidal import has its MODELED impact"

Response (10): "modeled" added. Now on Page 8 Line 22 in the "accept-change" version of the revised manuscript.

p8, line 19: ...bivalves that APPEAR TO exert..."

Response (11): This has been revised as suggested. Now on Page 8 Line 23 in the "accept-change" version of the revised manuscript.

p8, line 33: "a similar spatial gradient" implies that the reader can see the correspondence in space in Fig. 11. If you group chla values at the more coarse resolution of the imagery, do you see correlations in space? There may be a bias (y intercept) or proportional relationship (slope > or <1), but there should be correlation at the pixel scale if there is a similar spatial gradient. All that you would need to report is the r2 and p value in the text.

Response (12): We have calculated the correlation between satellite-measured and modeled chl-a. Their correlation is significant ($r^2 = 0.118$, $n = 2817$, $p < 0.0001$). These numbers have been added to the sentence. Now on Page 9 Line 4 in the "accept-change" version of the revised manuscript.

p9, line 7: put refs in chronological order

Response (13): Corrected. Now on Page 9 Lines 11–12 in the "accept-change" version of the revised manuscript.

p12, line 22: "It should be noted THAT the spatial..."

Response (14): Corrected. Now on Page 13 Line 30 in the "accept-change" version of the revised manuscript.

[revised manuscript text omitted]